# Scalable Membership Inference Attacks via Quantile Regression

**Martin Bertran** [*]
Amazon AWS AI/ML

**Shuai Tang** [*]
Amazon AWS AI/ML

**Michael Kearns**
University of Pennsylvania
Amazon AWS AI/ML

**Jamie Morgenstern**
University of Washington
Amazon AWS AI/ML

**Aaron Roth**
University of Pennsylvania
Amazon AWS AI/ML

**Zhiwei Steven Wu**
Carnegie Mellon University
Amazon AWS AI/ML

## Abstract

Membership inference attacks are designed to determine, using black box access to trained models, whether a particular example was used in training or not. Membership inference can be formalized as a hypothesis testing problem. The most effective existing attacks estimate the distribution of some test statistic (usually the model's confidence on the true label) on points that were (and were not) used in training by training many *shadow models*—i.e. models of the same architecture as the model being attacked, trained on a random subsample of data. While effective, these attacks are extremely computationally expensive, especially when the model under attack is large.

We introduce a new class of attacks based on performing quantile regression on the distribution of confidence scores induced by the model under attack on points that are not used in training. We show that our method is competitive with state-of-the-art shadow model attacks, while requiring substantially less compute because our attack requires training only a single model. Moreover, unlike shadow model attacks, our proposed attack does not require any knowledge of the architecture of the model under attack and is therefore truly "black-box". We show the efficacy of this approach in an extensive series of experiments on various datasets and model architectures. Our code is available at github.com/amazon-science/quantile-mia.

## 1 Introduction

The basic goal of privacy-preserving machine learning is to find models that are predictive on some underlying data distribution, without being disclosive of the particular data points on which they were trained. The simplest kind of attack that can be launched on a trained model—falsifying privacy guarantees—is a membership inference attack. A membership inference attack, informally, is a statistical test that is able to reliably determine whether a particular data point was included in the training set used to train the model or not.

Almost all membership inference attacks are based on the observation that models tend to overfit their training sets in different ways. In particular, they tend to systematically predict higher confidence in the true labels of data points from their training set, compared to points drawn from the same distribution not in their training set. The confidence that a model places on the true label of a data-point is thus a natural test statistic to build a membership-inference hypothesis test around. A

---

[0]Martin and Shuai are the lead authors, and other authors are ordered alphabetically. {maberlop,shuat}@amazon.com

37th Conference on Neural Information Processing Systems (NeurIPS 2023).

variety of recent methods [Shokri et al., 2017, Long et al., 2020, Sablayrolles et al., 2019, Song and Mittal, 2021, Carlini et al., 2022] are based around this idea, and aim to estimate the distribution of the test statistic (the confidence assigned to the true label of a datapoint) over the distribution of datapoints that were *not used in training* (and sometimes, also over the distribution of datapoints that were used in training) for the purpose of designing tests that can reject the null hypothesis—that a data point under attack was not used in training—with the desired level of confidence.

The efficacy of this class of attacks depends in large part on the granularity to which the distribution of the test statistic can be estimated. The simplest (and most computationally efficient) approach, originally proposed by Yeom et al. [2018], is to estimate this distribution *marginally* — i.e. without conditioning on the covariates $x$ of the example being attacked. This reduces the problem to a simple one-dimensional estimation problem, and—under mild assumptions—the optimal hypothesis test (by the Neyman-Pearson Lemma) is simply a fixed threshold $\tau$ on the test statistic—examples are declared to have been used in training if the confidence the model places on their true label exceeds $\tau$, and are declared to have been not used in training otherwise. More sophisticated methods attempt to estimate the distribution of the test statistic conditional on the inclusion of a target point $x$ in the training data (over the randomness of the selection of the other points used in training). Our method and others follow this approach, where the confidence score produced by each example $x$ by the target model is compared to a sample-dependent threshold $\tau(x)$— points $x$ with scores exceeding this threshold are declared to be used in training. The most common method under this approach is to train *shadow models* [Shokri et al., 2017, Long et al., 2020, Sablayrolles et al.,

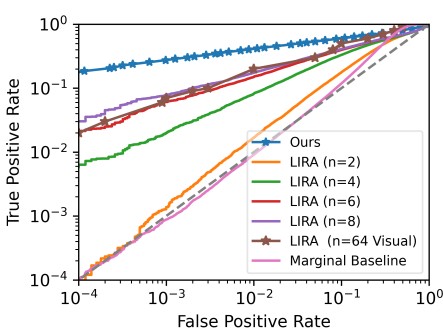

Figure 1: Comparing the true positive rate vs. false positive rate of our membership inference attack with the marginal baseline proposed in Yeom et al. [2018] and the state-of-the-art LiRA proposed in Carlini et al. [2022] evaluated at 2, 4, 6, and 8 shadow models. We also provide a visual readout of their 64 shadow model results, as reported in their paper (we did not have the compute necessary to reproduce this). We faithfully replicated LIRA's attack setup and produced better results than their reported values. Our single-model quantile regression attack can reliably identify training samples on a ResNet-50 ImageNet target model (67.5% test accuracy) without knowledge of the target architecture.

2019, Song and Mittal, 2021, Carlini et al., 2022]. Informally, shadow models are trained with the same architecture as the model being attacked, using random subsets of training data, that either include or do not include the target point $x$. As a result, each shadow model gives a sample of the test statistic conditional on $x$'s inclusion (or non-inclusion) in the training set, where the randomness is over the *other examples* in the training set and any randomness involved in training. Because many samples from this distribution on the test statistic are needed to estimate it, membership inference attacks based on shadow models generally require training many shadow models of the same architecture as the model under attack; between 64 and 256 shadow models were used in Carlini et al. [2022] (Figure 1 compares the receiver operating characteristic (ROC) of the attack on ImageNet against our proposed approach). Especially for large models, this makes shadow model attacks prohibitive, for at least two reasons:

1. **Training Cost**: Widely used commercial models, on which membership inference attacks would be most damaging, are extremely large and expensive to train. An attacker launching a membership inference attack based on shadow models must train many (dozens to hundreds) models of the same architecture. Thus the computational costs can be hundreds of times larger than the costs of training the model under attack, which for commercial models is prohibitive for attackers without enormous resources.

2. **Knowledge of the Model Under Attack:** As argued in Carlini et al. [2022], and also shown in Appendix A, when the shadow model is of the same complexity as or more complex than the target model, the LiRA attack performs well, and when less complex shadow models are deployed, the success rate of the attack drops precipitously. Hence the success of the attack depends on knowledge of the model under attack. But many aspects of the architecture and

training process for large commercial models are not publicly known, making this style of attack less effective in realistic settings.

## 1.1 Our Results

We introduce a new class of membership inference attacks, based on quantile regression, that is able to mitigate these issues:

1. Like attacks based on shadow models, it is a *conditional* attack, subjecting different examples $x$ to different thresholds $\tau(x)$. However,

2. It only requires training a single model, and

3. The architecture of the model used in the attack need not be related to the architecture of the model under attack, and so no knowledge of the model architecture or training algorithm used to train the model under attack is needed.

**Our quantile regression attack**. Given a model $f$ that we intend to attack, we collect a dataset of labelled examples $\{(x_i, y_i)\}_{i=1}^n$ from the underlying data distribution, known to not have been used in training.[1] For each example $(x_i, y_i)$ in our dataset, we evaluate the model $f$ on example $x_i$, and record the (real-valued) confidence score $s(x_i, y_i)$ that the model places on the correct label $y$. We then train a quantile regression model $q$ on the dataset $\{(x_i, s(x_i, y_i))\}$ consisting of examples $x$ labeled with their confidence scores $s(x, y)$. Informally, the quantile regression model is trained to predict $q(x)$, a target $1 - \alpha$ quantile of the conditional distribution on $s(x_i, y_i)$, given $x_i$.[2] Intuitively, a score $s(x_i, y_i)$ larger than the $1 - \alpha$ quantile $q(x_i)$ indicates that $f$ assigns a confidence on the true label that is higher than a $1 - \alpha$ fraction of the examples not used in training — giving us evidence that the example in question was part of the training set. Thus, given a new target point $(x, y)$, we *reject the null hypothesis* that $x$ was not used in training (i.e. we declare $x$ to have been used in training) whenever $s(x, y)$ exceeds $q(x)$ — i.e. when $s(x, y) \geq q(x)$. Similarly, whenever $s(x, y) < q(x)$, we do not reject the null hypothesis, and declare the point $(x, y)$ to have not been used in training.

This attack by design has a false positive rate of $\alpha$— the probability that it incorrectly declares a randomly selected point $(x, y)$ that was not used in training to have been used in training is $\alpha$. The ability of $q$ to correctly label those examples used in training as such, measured by its true positive rate or precision or related statistics, will vary with $\alpha$ (the higher $\alpha$, the larger the number of positive labels our test will assign). So, in varying our target $\alpha$, we can sweep out our test's tradeoff between false positive and true positive rates.

The primary strength of our attack is that we need only a *single* quantile regression model $q$, rather than a large number of shadow models. Furthermore, because the success of our attack depends only on how well $q$ predicts the quantiles of the confidence score distribution of $f$ (rather than producing confidence scores drawn from the same distribution as $f$), $q$ need not have any relationship to the architecture of $f$ or any knowledge of it— the only access to $f$ that is needed is the ability to evaluate confidence scores $s(x, y)$ produced by $f$ given examples $x$. Our attack is, therefore, more "black-box" than those which use shadow models of the same architecture as $f$.

We derive a basic theory for our approach based on quantile regression, which trains a model to predict quantiles by minimizing *pinball loss*. We run an extensive series of experiments and find that our quantile regression approach is competitive with (and sometimes more effective than) much more computationally expensive shadow model approaches. The relative effectiveness of our approach appears to grow the more complex the classification task and model under attack are. For example, when attacking a ResNet-50 model trained on ImageNet-1k, our attack (which trains only a single model) outperforms shadow model approaches trained much more expensively at every false positive rate. On simpler and less data rich tasks (like CIFAR-10), the accuracy of our approach dominates the marginal baseline of Yeom et al. [2018], but falls short of shadow model approaches. Thought provokingly, however, we find that when this occurs, it is because the shadow model approach has

---

[1]Under the presumption that only a small fraction of data sampled from the distribution were used in training, then we may simply take a random sample from the underlying distribution, and be confident that it is representative of data not used in the training procedure for $f$.

[2]This is an informal description, as in realizable settings, conditioning on $x_i$ in its entirety leaves a point mass distribution on $s(x_i, y_i)$ — i.e. the deterministic confidence score for $y_i$ predicted by the model $f(x_i)$. See Section 3 for precise guarantees.

found thresholds that correspond to a quantile model with lower pinball loss than our trained quantile regression model. This suggests that our fundamental approach of pinball loss minimization is sound, and that our attempts to directly optimize for it are less successful when data is less plentiful. Across all experiments, we find that the best quantile regression method (as measured by pinball loss) is uniformly the best membership inference attack.

## 1.2 Additional Related Work

Starting with the seminal work of Homer et al. [2008], membership inference has become one of the most widely studied classes of privacy attacks. Most approaches for membership inference determine whether an example is part of the training set via some score function, which can be loss [Yeom et al., 2018, Sablayrolles et al., 2019], confidence [Salem et al., 2018], entropy [Song and Mittal, 2021], or difficulty calibration [Watson et al., 2021] among others. Another common approach is to query the model on similar or related examples to the target point [Wen et al., 2023, Jayaraman et al., 2020, Long et al., 2018, Li and Zhang, 2020].

We focus the remainder of our discussion on related work on shadow model approaches to membership inference since they are our main benchmarks. Most work on shadow models considers a setup where there is a private dataset $D^{\text{private}}$ (unknown to the attacker) drawn from a distribution $Q$, and an algorithm $\mathcal{A}$ for training a model $f = \mathcal{A}(D^{\text{private}})$. The attacker has access to a set of data $D^{\text{public}}$ drawn from the same distribution $Q$ and partial query access to the model $f$ from which the attacker can compute scores $s(x, y)$ given target examples $(x, y)$. The attacker aims to predict, for a data point $(x, y)$, whether $(x, y) \in D^{\text{private}}$.

**Likelihood Ratio Attacks.** Membership inference attacks are fundamentally hypotheses tests between two competing hypotheses ($H_0 : (x, y) \notin D^{\text{private}}$, $H_1 : (x, y) \in D^{\text{private}}$). By the Neyman-Pearson lemma [Neyman and Pearson, 1933], the optimal hypothesis test based on a test statistic $s(x, y)$ computes the likelihood ratio of the score under the null and alternative hypothesis, and subjects the likelihood ratio to a threshold $\tau$. The choice of the threshold $\tau$ determines the trade-off between precision (the fraction of examples labeled as belonging to the private dataset which did belong to the dataset) and recall (or true positive rate) of the resulting classifier. We call a membership inference attack carried out with this classifier a likelihood ratio attack (LiRA), introduced by Carlini et al. [2022]. LiRA was designed to achieve very high precision (very few false positives relative to the number of positive predictions), as they noted high precision corresponds to a high degree of confidence that the data points accused of being part of the training set were, in fact, part of the training set. Prior work had looked at global notions of inference attack quality, at possibly much lower degrees of precision [Ye et al., 2021, Jayaraman et al., 2021].

The main difficulty with implementing LiRA directly is that the density functions of the score under the null and alternative hypothesis are unknown. Instead, the literature aims to estimate these density functions, primarily by training a collection of *shadow* models [Shokri et al., 2017, Long et al., 2020, Sablayrolles et al., 2019, Song and Mittal, 2021, Carlini et al., 2022]. Shadow model attacks split the attacker's dataset $D^{\text{public}}$ into several pairs of shadow public/private datasets $D_i^{\text{private}}, D_i^{\text{public}}$, and for each of these shadow datasets, a shadow model $f_i$ is trained on $D_i^{\text{private}}$. The shadow model $f_i$, and corresponding datasets $D_i^{\text{private}}, D_i^{\text{public}}$ are used to generate private and public score samples $s_i^{\text{private}}, s_i^{\text{public}}$ from which to estimate the likelihood ratio function given parametric assumptions. Carlini et al. [2022] used a large number of shadow models to achieve high precision. This approach works well—but it is computationally demanding because it requires training many shadow models.

## 2 Preliminaries

We study attacks on models $f$ that solve a supervised learning problem defined over a distribution $\mathcal{D} \in \Delta(\mathcal{X} \times \mathcal{Y})$ of labeled examples $(x, y)$, consisting of *feature vectors* $x \in \mathcal{X}$ and labels $y \in \mathcal{Y}$. We make no assumptions about $\mathcal{X}$ or $\mathcal{Y}$ (e.g. $\mathcal{Y}$ could be a discrete set in a multi-class classification problem, or we could have $\mathcal{Y} = \mathbb{R}$ in a regression problem). We assume that the model $f$ outputs a *confidence score* in $[0, 1]$ for each possible label $\hat{y} \in \mathcal{Y}$: in other words, $f : \mathcal{X} \to [0, 1]^{\mathcal{Y}}$, and for each $\hat{y} \in \mathcal{Y}$, we write $f_{\hat{y}}(x) \in [0, 1]$ for the confidence score that $f$ assigns to label $\hat{y}$ given input $x$. Such models are often used to make point predictions by predicting the label $\hat{y} = \arg\max_{y \in \mathcal{Y}} f_y(x)$ on input $x$ — but we will interact with such models $f$ only at the level of confidence score predictions.

A model $f$ is derived from a *training process* that did not have direct access to $\mathcal{D}$, but rather to a finite sample $D^{\text{private}}$ called the training set. The training process correlates $f$ with $D^{\text{private}}$. A membership inference attack is a hypothesis test that must use a test statistic derived only from $f$ that aims to determine whether a labeled example $(x, y)$ is a member of the training set $D^{\text{private}}$ or not. Formally, we model this as a hypothesis test that aims to solve the following simple hypothesis testing problem:

$$H_0 : (x, y) \sim \mathcal{D} \qquad H_1 : (x, y) \sim D^{\text{private}}$$

Here, $(x, y) \sim D^{\text{private}}$ denotes sampling a point $(x, y)$ uniformly at random from $D^{\text{private}}$. Observe that since we derive the test statistic from $f$, even if $D^{\text{private}}$ was itself sampled i.i.d. from $\mathcal{D}$, $H_0$ and $H_1$ are distinct hypotheses since the training process has correlated $f$ with $D^{\text{private}}$. In particular, we will base our attack on the presumption that $f$ will tend to be over-confident on examples $(x, y) \in D^{\text{private}}$. Towards this, we choose as our test statistic $s(x, y) = z_y(x) - \max_{y' \neq y} z_{y'}(x)$ the logit difference between the true label and its most likely alternative, where $z(x)$ denotes the logits (unnormalized features before the softmax nonlinearity) of the model. This choice follows the scoring rule used in Carlini et al. [2022] and will be useful for experimental comparisons. However, the remainder of our theoretical treatment will be agnostic as to this choice.

In this paper we restrict attention to membership inference attacks (hypothesis tests) that apply a threshold function to $s(x, y)$, with a threshold that may depend on $x$. Given a function $q : \mathcal{X} \to \mathbb{R}$ that maps examples $x$ to thresholds, the corresponding membership inference attack is given by:

$$\mathcal{A}_q(x, y) = \begin{cases} \top \;\; (x \sim \mathcal{D}) & \text{if } s(x, y) < q(x) \\ \bot \;\; (x \sim D^{\text{private}}) & \text{if } s(x, y) \geq q(x). \end{cases}$$

Here $\bot$ is shorthand for "We reject the null hypothesis that $(x, y) \sim \mathcal{D}$ (and thus declare $(x, y)$ of being in the training set)", and $\top$ is shorthand for "we do not reject the null hypothesis (and thus do not accuse $(x, y)$ of being in the training set)".

A natural baseline is to set $q(x, y) = \tau$ to be a constant. This is the attack proposed by Yeom et al. [2018], and we write this baseline as $\mathcal{A}_\tau$. If $\tau$ is set to be a $1 - \alpha$ quantile of the marginal distribution on $s(x, y)$ when $(x, y) \sim \mathcal{D}$, then this attack can easily be seen to have *false positive rate* $\alpha$. Below we define quantiles assuming (for simplicity) that the distribution in question is continuous—but it is also possible to define quantiles without this assumption.

**Definition 1.** *Fix a continuous distribution $\mathcal{P} \in \Delta\mathbb{R}$. A number $\tau \in \mathbb{R}$ is a $(1 - \alpha)$-quantile of $\mathcal{P}$ if:*

$$\Pr_{s \sim \mathcal{P}}[s \leq \tau] = 1 - \alpha$$

We can evaluate the performance of a membership inference attack by evaluating its false positive rate, true positive rates, and precision[3]:

**Definition 2.** *Fix an arbitrary membership inference attack $\mathcal{A} : \mathcal{X} \times \mathcal{Y} \to \{\top, \bot\}$. We define the following performance metrics*

$$FPR(\mathcal{A}) = \Pr_{(x,y) \sim \mathcal{D}}[\mathcal{A}(x, y) = \bot], \qquad TPR(\mathcal{A}) = \Pr_{(x,y) \sim D^{private}}[\mathcal{A}(x, y) = \bot],$$

$$Prec(\mathcal{A}) = \frac{FPR(\mathcal{A})}{FPR(\mathcal{A}) + TPR(\mathcal{A})}.$$

It is immediate that the baseline membership inference attack achieves its target false positive rate; the true positive rate and precision of the attack can be evaluated empirically:

**Lemma 1.** *Let $\tau$ be a $1 - \alpha$ quantile of $\mathcal{P}$, the distribution on confidence scores $s(x, y)$ that results from sampling $(x, y) \sim \mathcal{D}$. Then the baseline membership inference attack $\mathcal{A}_\tau$ has $FPR(\mathcal{A}_\tau) = \alpha$.*

*Proof.* This follows from the definitions: $FPR(\mathcal{A}_\tau) = \Pr_{(x,y) \sim \mathcal{D}}[s(x, y) \geq \tau] = \alpha$. $\qquad \square$

---

[3]Precision is equivalent to the accuracy of the attack conditioned on a positive prediction $\bot$ when $\Pr[\bot] = 0.5$

## 3   Our Attack

Our attack is $\mathcal{A}_q(x, y)$, where $q$ is derived from a *quantile regression* model trained to predict quantiles of our test statistic $s(x, y)$ on a dataset of points $(x, y)$ drawn from our null hypothesis distribution $(x, y) \sim \mathcal{D}$. A popular non-parametric quantile regression method is to minimize *pinball loss*, which elicits *quantiles* (just as squared loss elicits means):

**Definition 3.** *The pinball loss function defined for a $1 - \alpha$ quantile is:*

$$PB_{1-\alpha}(\hat{y}, y) = \max\{\alpha(\hat{y} - y), (1 - \alpha)(y - \hat{y})\}$$

Pinball loss is a useful objective function because it elicits quantiles:

**Lemma 2.** *Fix any distribution $\mathcal{P} \in \Delta\mathbb{R}$. Let:*

$$\tau \in \arg\min_{\hat{y} \in \mathbb{R}} \mathbb{E}_{y \sim \mathcal{P}}[PB_{1-\alpha}(\hat{y}, y)]$$

*Then $\tau$ is a $(1 - \alpha)$-quantile of $\mathcal{P}$.*

Viewed through this lens, the baseline attack can be thought of as the end result of the following simple pipeline:

1. Select a target false positive rate $\alpha$,
2. Choose a threshold $\tau$ by solving the minimization problem

$$\tau \in \arg\min_{\tau' \in \mathbb{R}} \mathbb{E}_{(x,y) \sim \mathcal{D}}[PB_{1-\alpha}(\tau', s(x, y))]$$

3. Instantiate the baseline membership inference attack $\mathcal{A}_\tau$.

Our attack departs from this baseline attack by training a model $q : \mathcal{X} \to \mathbb{R}$ on feature/confidence score pairs to optimize pinball loss, rather than a single threshold $\tau$:

1. Select a target false positive rate $\alpha$ and a class of model architectures $\mathcal{H}$ consisting of models $q : \mathcal{X} \to \mathbb{R}$.
2. Train a model $q \in \mathcal{H}$ by solving the following risk minimization problem:

$$q \in \arg\min_{q' \in \mathcal{H}} \mathbb{E}_{(x,y) \sim \mathcal{D}}[PB_{1-\alpha}(q(x), s(x, y))] \tag{1}$$

3. Instantiate the membership inference attack $\mathcal{A}_q$

We train our quantile regression model on a dataset consisting of points $(x, y)$ drawn from the underlying distribution (of points *not* used in training), labeled by the confidence scores $s(x, y)$ derived from the model. Thus our attack assumes only that we have API access to the model under attack $f$, and are able to query it on a finite set of points to obtain confidence scores.

We now establish some basic properties of our attack. The first is that, like the baseline attack, it actually achieves its target false positive rate. Unlike the baseline attack, this is no longer immediate, but can be derived from properties of the pinball loss:

**Theorem 1.** *Fix a distribution $\mathcal{D} \in \Delta(\mathcal{X} \times \mathcal{Y})$ over labeled examples and a model $f$. Suppose that the marginal distribution over $s(x, y)$ for $(x, y) \sim \mathcal{D}$ is continuous. Let $\mathcal{H}$ be any class of models that is closed under additive shifts — i.e. such that for each $q \in \mathcal{H}$ and $\Delta \in \mathbb{R}$, then we also have $q' \in \mathcal{H}$ for $q'(x) = q(x) + \Delta$. Then for the membership inference attack $\mathcal{A}_q$ produced by our method, $FPR(\mathcal{A}_q) = \alpha$.*

We defer the proof to Appendix C.1. Thus by varying $\alpha$, we can use our attack to sweep out a curve trading off our target false positive rate with our (empirically measured) true positive rate, just as we can for the baseline attack $\mathcal{A}_\tau$.

Is this guarantee stronger than the baseline attack, and if so, in what sense? To give one perspective on this, it will be helpful to define group conditional quantile consistency, which is related to multicalibration, a concept originating from the fairness in machine learning literature [Hébert-Johnson et al., 2018, Gupta et al., 2022, Bastani et al., 2022, Jung et al., 2023, Noarov and Roth, 2023].

**Definition 4.** *Fix a collection $\mathcal{G}$ of group indicator functions $g : \mathcal{X} \to \{0, 1\}$ and a model $q : \mathcal{X} \to \mathbb{R}$. $q$ satisfies group conditional quantile consistency with respect to a distribution $\mathcal{P} \in \Delta(\mathcal{X} \times \mathbb{R})$, a target quantile $1 - \alpha$, and the collection of groups $\mathcal{G}$ if for every $g \in \mathcal{G}$:*

$$\Pr_{(x,s)\sim\mathcal{P}}[q(x) \leq s | g(x) = 1] = 1 - \alpha$$

Group conditional quantile consistency asks that our quantile predictions be correct not just marginally over the data, but also (simultaneously) conditionally on membership in a large number of potentially intersecting groups. If we optimize pinball loss over a richer set of models (that are closed under shifts by a class of group indicator functions, rather than just constant functions), then our attack will achieve its target false positive rate even when conditioning on membership in each of the groups specified by the functions $g \in \mathcal{G}$, rather than just marginally. This is a stronger guarantee, as a marginal false positive rate on its own need not hold subject to additional conditioning events.

**Theorem 2.** *Fix a distribution $\mathcal{D} \in \Delta(\mathcal{X} \times \mathcal{Y})$ over labeled examples and a model $f$. Fix a collection of group indicator functions $\mathcal{G}$ and a class of models $\mathcal{H}$ such that:*

1. *$\mathcal{H}$ is closed under shifts from $\mathcal{G}$: for every $h \in \mathcal{H}$, $g \in \mathcal{G}$, and $\eta \in \mathbb{R}$, the function $h'(x) = h(x) + \eta g(x)$ is such that $h' \in \mathcal{H}$.*

2. *The conditional distribution over $s(x, y)$ for $(x, y) \sim \mathcal{D}$, conditional on $g(x) = 1$ is continuous for all $g \in \mathcal{G}$.*

*Then for the membership inference attack $\mathcal{A}_q$ produced by our method, its false positive rate is $1 - \alpha$ conditional on membership in each group $g \in \mathcal{G}$: $\Pr_{(x,y)\sim\mathcal{D}}[\mathcal{A}_q(x, y) = \perp | g(x) = 1] = 1 - \alpha$.*

We defer the proof to Appendix C.2.

# 4 Experiments

We present two sets of experiments on two different data domains, including images and tabular data. Here, we mainly focus on attacking widely-used classification models in these two domains, however, our theoretical claims generalizes to other data domains as well.

## 4.1 Image Classification Experiments

We evaluate the effectiveness of our proposed approach on four image classification datasets: CIFAR-10 [Krizhevsky et al., 2009], a standard image classification dataset with 10 target classes; CIFAR-100 [Krizhevsky et al., 2009] another image classification dataset with 100 target classes; ImageNet-1k [Russakovsky et al., 2015], a substantially larger image classification task with 1000 target classes; and CINIC-10 [Darlow et al., 2018], an extension of CIFAR-10 that additionally uses images from ImageNet-1k corresponding to the original 10 target classes. To provide a realistic evaluation, we ensure our base models use common, well-performing architectures and follow standard guidelines for hyperparameter selection [He et al., 2015], including data augmentation, learning rate schedule, and l2 regularization(weight decay). For CIFAR-10 and CIFAR-100, target (classification) models include ResNet-10, ResNet-18, ResNet-34, and ResNet-50 [He et al., 2015]. For ImageNet-1k and CINIC-10, the target model is a ResNet-50. In all experiments, 50% of the dataset is used for training the target model, and, following the common standards, the resolution of the target model is 32x32 for CIFAR and CINIC datasets, and 224x224 for the ImageNet-1k dataset. The accuracy of each target model is presented in Appendix B.

To perform our membership inference attack, we train a single quantile regression model following our proposal in Eq.(1). One of the advantages of our attack is that it is model-agnostic: since it does not require the knowledge of the model architecture of the target or the knowledge of the training algorithm, we use the same model architecture for our quantile regression model in all settings : a pretrained ConvNext-Tiny model [Liu et al., 2022]. On CINIC-10 we additionally experimented with a ResNet-50 model as our quantile model architecture.

For the scoring function, we use the hinge score proposed in Carlini et al. [2022] $s_{hinge}(x, y) = z_y(x) - \max_{y' \neq y} z_{y'}(x)$. This scoring rule is closely related to the logit function of the model's confidence $s(x, y) = \log(f_y(x)) - \log(1 - f_y(x))$ for models with high label confidence, and has

been empirically shown to be approximately normally distributed. On the largest dataset, ImageNet-1k, we directly train a quantile regression model to predict (log-spaced) quantiles at $\alpha \in [0.9996, 1]$ An alternative is to learn a parameterized model so that, for each sample, the model predicts the mean $\mu(x)$ and the log of the standard deviation $\log \sigma(x)$, and the quantiles of a sample can be generated from the Gaussian distribution $\mathcal{N}(\mu(x), (e^{\log \sigma(x)})^2)$. Due to the fact that CIFAR and CINIC datasets are much smaller (25000 samples available for training on CIFAR), rather than directly learning quantiles, we opt to learn the parametrized models with Gaussian parameters on these datasets.

All images are processed at 224x224 resolution by the quantile model, which is trained on the remaining 50% of the training samples that were not used to train the target model. Since there is a smaller body of literature on stable hyperparameters for regression models, we use Ray Tune [Liaw et al., 2018] for hyperparameter tuning (tuning is used to minimize validation pinball loss in a held out dataset). The compute budget for our attack was approximately 30 GPU minutes per quantile regression attack (4 hours including hyperparameter optimization) on CIFAR-10/CIFAR-100, 18 minutes (4 hours 40 minutes including hyperparameter optimization) on CINIC-10, and 16 hours (128 hours including hyperparameter optimization) on ImageNet-1k. Final hyperparameters were found to be consistent across all tasks sharing an architecture; more information is provided in Appendix D

Figures 1 and 2 shows ROC curves of our proposed approach on ImageNet-1k and CINIC-10 respectively; FPR is computed on a held-out dataset that was not used to train the target or the quantile regression model. Both the marginal quantile approach from Yeom et al. [2018] and the shadow model approach LIRA from Carlini et al. [2022] are also shown for reference. In these experiments, our quantile regression approach dominates the shadow model approach at all comparison points for ImageNet-1k, and is roughly comparable to 4 shadow models on CINIC-10. Appendix E shows the same comparison against all CIFAR-10/100 target models (ResNet-10, ResNet18, ResNet34, ResNet-50). In these latter experiments, our attack outperforms the marginal baseline but falls short of the shadow model approach. We note that in this case, the shadow models actually produce thresholds that have lower pinball loss than our quantile regression algorithm, suggesting that on the smaller CIFAR datasets, our optimization heuristic was unable to sufficiently minimize test pinball loss. We find this observation to be consistent across experiments, that the attack with smaller pinball loss on public data is generally the attack with the best TPR at that FPR level $\alpha$

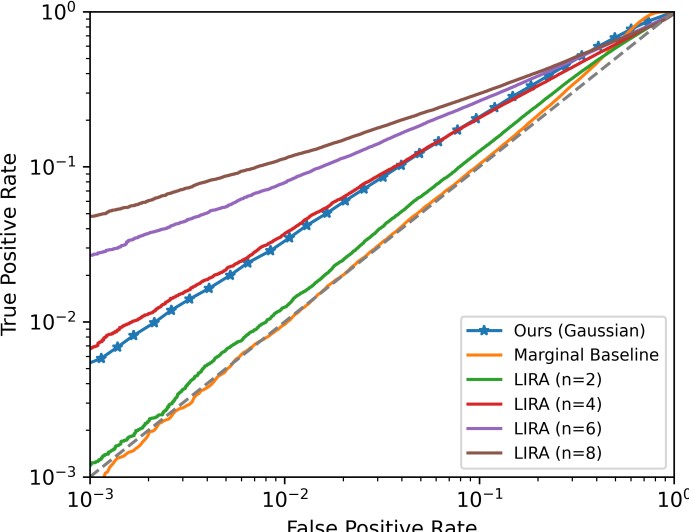

Figure 2: Comparing the true positive rate vs. false positive rate of our membership inference attack (with parametric Gaussian loss) on the CINIC-10 dataset (an extension of the CIFAR-10 dataset with 270,000 images, 4.5 times that of CIFAR-10) with the marginal baseline proposed by Yeom et al. [2018] and the state-of-the-art LiRA proposed in Carlini et al. [2022] evaluated at 2, 4, 6, and 8 shadow models. Both the model under attack and our quantile regression model use a ResNet-50 architecture. Both LiRA and our method are able to reliably identify train samples at very low false positive rates.

Table 1: Precision of all membership inference attack at 1% and 0.1% false positive rates on ResNet-50 architectures. Our attack consistently dominates the marginal baseline and produces excellent results on ImageNet-1k compared to shadow model approaches. We do not beat the shadow model approach on the smaller CIFAR datasets, potentially owing to their small dataset size. On CINIC-10, a larger dataset than CIFAR, we obtain comparable results to 4 shadow models. Additional results for the remaining architectures are presented in Appendix E

| | Precision @ $1\%$ FPR | | | | Precision @ $0.1\%$ FPR | | | |
|---|---|---|---|---|---|---|---|---|
| Method | C-10 | C-100 | CIN10 | IN-1k | C-10 | C-100 | CIN10 | IN-1k |
| Marginal | 48.56% | 58.81% | 49.02% | 47.62% | 60.94% | 65.75% | 45.76% | 46.81% |
| LIRA (n=2) | 78.55% | 95.21% | 55.43% | 62.70% | 83.18% | 98.65% | 54.07% | 56.04% |
| LIRA (n=4) | 80.52% | 95.87% | 78.71% | 89.11% | 91.48% | 98.94% | 86.99% | 95.18% |
| LIRA (n=6) | 83.19% | 96.20% | 88.75% | 93.74% | 93.17% | 99.02% | 96.40% | 98.38% |
| LIRA (n=8) | 83.00% | 96.07% | 91.86% | 94.57% | 93.70% | 98.98% | 97.94% | 98.73% |
| Ours | 62.95% | 79.57% | 76.67% | 97.45% | 64.48% | 85.41% | 85.46% | 99.64% |

Table 1 shows precision of the proposed membership inference attack at 1% and 0.1% false positive rate on the ResNet-50 target networks (additional results shown in Appendix E). Our attack robustly predicts membership in the private dataset with high precision even at low FPR. The proposed approach works on all target architectures and datasets, but works particularly well on the more complex and data intensive ImageNet-1k task.

Since LIRA produces an explicit score distribution $\mathcal{N}(s(x,y); \mu(x,y), \sigma(x,y))$ based on the shadow model's predictions, we can compare all 3 methods (Ours, LIRA, marginal baseline) in terms of pinball loss on public data ($x \sim \mathcal{D}$). **We find that the attack with the smallest pinball loss on public data is the better membership inference attack across all datasets.** This shows that a strong quantile predictor on public data is a strong membership inference attack; which validates the core premise of our approach in Section 3.

A possible explanation for the relative lack of success of directly learning a quantile regression model on CIFAR datasets could be the relatively low number of available samples (25000). In such low data scenarios, training shadow models is also computationally affordable. The opposite holds true for the much larger ImageNet1k dataset, on which the computational cost of training a single target model, much less multiple shadow models far exceeds the cost of a single quantile regression model.

## 4.2 Tabular Classification Experiments

In addition to experiments on image datasets, we here demonstrate the effectiveness of our membership inference attack on tabular datasets, including large datasets from derived from the US Census' American Community Survey (ACS) [Ding et al., 2021] and small ones from OpenML [4] [Grinsztajn et al., 2022]. Gradient boosting with decision trees is widely-used for classification tasks with tabular data, so in our experiments, to achieve a reasonable performance, we train a gradient boosting model with 5-fold cross validation for hyperparameter tuning on the private portion of the data. For our attack model, gradient boosting with regression trees is applied, but now with regression targets as mentioned above. Hyperparameters for regression tasks are also tuned using a public portion of the data to avoid overfitting. In our experiments, catboost is used for model training, and Optuna [Akiba et al., 2019] is used for hyperparameter tuning.

Table 2 shows that our attack, which involves learning a single regression model, performs on par with the LiRA attack, which requires learning at least 16 models on some tasks and more on other tasks. Since each model, including our regression model and a shadow model in LiRA, has the same latency in terms of hyperparameter tuning and model training, our attack requires significantly less compute (equivalent to a single shadow model), and it reduces a successful attack from training many models to only one model. Figure 3 shows ROC curves of our proposed approach on OpenML.

## 5 Discussion

We have introduced a new family of membership inference attacks that are competitive with the state of the art (and in our ImageNet experiments, substantially and uniformly better), while requiring

---

[4]https://www.openml.org

Table 2: Precision @ 1% FPR on OpenML datasets and 0.5% on ACS datasets. Evaluating at different FPR levels is due to the fact that OpenML datasets have around 5,000 samples, and ACS datasets have around 20,000 samples. $n$ here denotes the number of shadow models trained for LIRA.

| | Precision @ 1% FPR (OpenML) | | | | Precision @ 0.5% FPR (ACS NY) | | | |
|---|---|---|---|---|---|---|---|---|
| Task ID | 361057 | 361064 | 361067 | 361070 | Coverage | Income | Travel | Mobility |
| LIRA (n=16) | 70.33% | 85.44% | 85.52% | 73.33% | 66.07% | 50.71% | 66.07% | 72.74% |
| LIRA (n=32) | 76.22% | 88.52% | 89.62% | 78.35% | 66.28% | 52.53% | 67.52% | 68.84% |
| LIRA (n=64) | 82.73% | 90.31% | 89.46% | 79.57% | 69.23% | 50.24% | 65.64% | 65.26% |
| Ours | 83.35% | 88.05% | 87.35% | 86.54% | 67.31% | 56.35% | 63.98% | 85.27% |

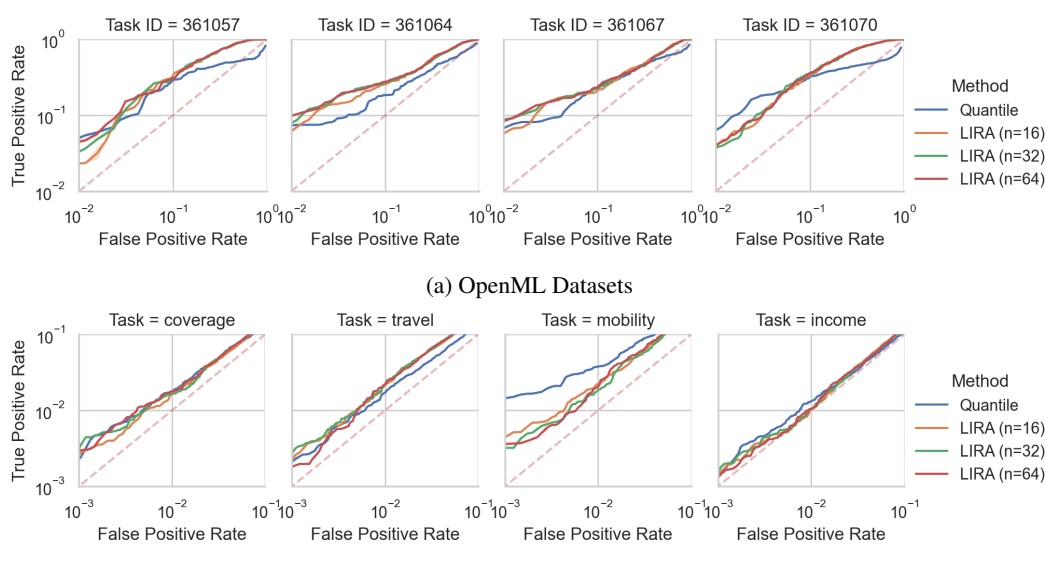

(a) OpenML Datasets

(b) ACS Task on NY data

Figure 3: Comparing the true positive rate vs. false positive rate of our membership inference attack against LIRA evaluated at 16, 32, and 64 shadow models on OpenML and ACS datasets. A single quantile regression model can produce similar results as multiple shadow models at a fraction of the compute cost for gradient boosting models.

substantially fewer computational resources and less knowledge of the target model. Moreover, we have identified pinball loss as a key target objective: uniformly across all of our experiments, the methods that produce thresholds minimizing pinball loss are the most effective attacks. Together, this brings membership inference closer to practicality on large commercial models. This serves to highlight a growing risk to privacy—but also provides a more efficient means to audit models by subjecting them to our attacks. We hope that our methods encourage and enable a more widespread practice of auditing models for privacy violations by subjecting them to membership inference attacks before deployment.

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

## A Shadow Model Architecture Mismatch

Here we explore how a shadow model membership inference attack is affected by the lack of knowledge of the target model. For this, we vary the shadow model's and target model's architecture between 6 different configurations: 3 CNN models with 32 filters each and varying number of pooling layers between 1 and 3, 4 Wide Residual Networks Zagoruyko and Komodakis [2016] of depth 28 and varying width ($[1, 2, 5, 10]$). We train a single target model and 4 shadow models; all models are trained with SGD with momentum and random augmentations. The results are shown in Figure 4, where small architecture mismatches (i.e, same model family of target and test architecture) generally degrade performance very little, but larger architecture mismatches can cause significant performance degradation.

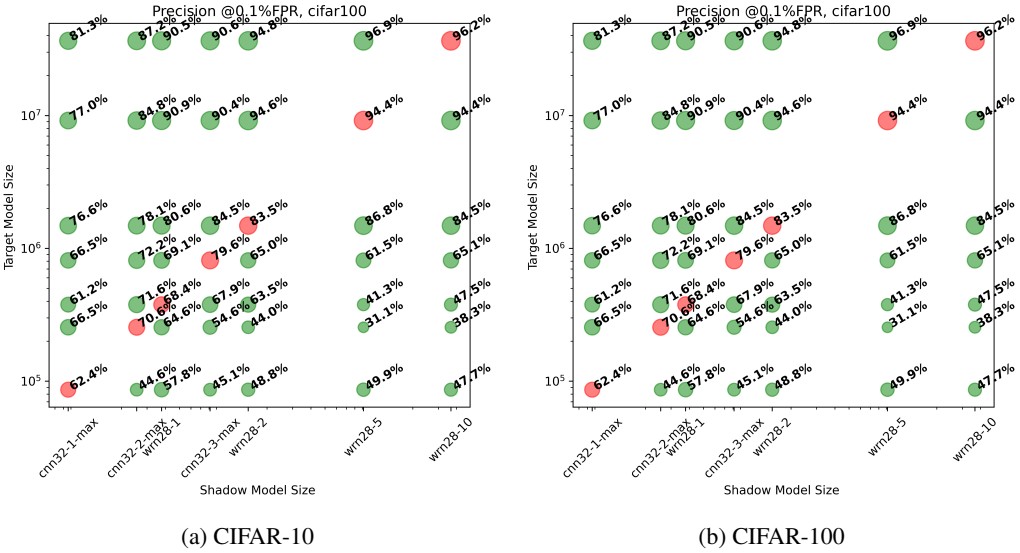

(a) CIFAR-10          (b) CIFAR-100

Figure 4: Comparing the precision at $0.1\%$ false positive rate of the LIRA attack on CIFAR-10 and CIFAR-100 on mismatched target and shadow model architectures. Red circles are used to denote scenarios where target and shadow model architectures match.

## B Target Model Accuracies

Here we summarize the accuracy of all target model architectures.

| | Architecture | | | |
|---|---|---|---|---|
| **Dataset** | **ResNet-10 Acc** | **ResNet-18 Acc** | **ResNet-34 Acc** | **ResNet-50 Acc** |
| CIFAR-10 | 90.3% | 90.5% | 90.7% | 91.0% |
| CIFAR-100 | 66.7% | 68.6% | 68.3% | 68.6% |
| ImageNet-1k | - | - | - | 67.5% |

Table 3: Accuracy of target classifiers. All target classifiers are trained on $50\%$ of the usual training data split using the training setup described in He et al. [2015]. Due to the reduced ammount of training data, these target networks have lower accuracies than those initially reported in He et al. [2015].

## C Proofs

### C.1 Proof of Theorem 1

*Proof.* To establish this theorem, the following lemma will be useful, which informally states that if we can shift a model's quantile predictions to get its "marginal coverage rate" to be closer to the target $1 - \alpha$, then we will also decrease the pinball loss of that model:

**Lemma 3** (Jung et al. [2023], Roth [2022])**.** *Fix any continuous distribution $\mathcal{P} \in \Delta\mathbb{R}$ with density bounded by $\rho$. Suppose $\hat{q}$ is a model such that $\Pr_{y \sim \mathcal{P}}[y \le \hat{q}(x)] = 1 - \alpha'$, and let $\Delta \in \mathbb{R}$ be such that $\Pr_{y \sim \mathcal{P}}[y \le \hat{q}(x) + \Delta] = 1 - \alpha$. Let $q(x) = \hat{q}(x) + \Delta$. Then:*

$$\mathop{\mathbb{E}}_{y \sim \mathcal{P}}[PB_{1-\alpha}(\hat{q}(x), y)] - \mathop{\mathbb{E}}_{y \sim \mathcal{P}}[PB_{1-\alpha}(q(x), y)] \ge \frac{(\alpha - \alpha')^2}{2\rho}$$

Recall that $q$ is chosen such that:

$$q \in \arg\min_{q' \in \mathcal{H}} \mathop{\mathbb{E}}_{(x,y) \sim \mathcal{D}}[\text{PB}_{1-\alpha}(q(x), s(x,y))]$$

For point of contradiction, suppose that $\text{FPR}(\mathcal{A}_q) = \alpha'$ for some $\alpha' \ne \alpha$. Expanding out the definition of the false positive rate, we have that:

$$
\begin{aligned}
\alpha' &= \text{FPR}(\mathcal{A}_q) \\
&= \mathop{\Pr}_{(x,y) \sim \mathcal{D}}[\mathcal{A}_q(x,y) = \bot] \\
&= \mathop{\Pr}_{(x,y) \sim \mathcal{D}}[s(x,y) \ge q(x)] \\
&= 1 - \mathop{\Pr}_{(x,y) \sim \mathcal{D}}[s(x,y) \le q(x)]
\end{aligned}
$$

So $\Pr_{(x,y) \sim \mathcal{D}}[s(x,y) \le q(x)] = 1 - \alpha'$. Let $\Delta \in \mathbb{R}$ be such that $\Pr_{(x,y) \sim \mathcal{D}}[s(x,y) \le q(x) + \Delta] = 1 - \alpha$—Note that such a $\Delta$ is guaranteed to exist by continuity of the distribution on $s(x,y)$. Let $q'(x) = q(x) + \Delta$. By Lemma 3,

$$\mathop{\mathbb{E}}_{(x,y) \sim \mathcal{D}}[\text{PB}_{1-\alpha}(q'(x), s(x,y))] < \mathop{\mathbb{E}}_{(x,y) \sim \mathcal{D}}[\text{PB}_{1-\alpha}(q(x), s(x,y))]$$

But because $\mathcal{H}$ is closed under additive shifts, we also have that $q' \in \mathcal{H}$. Together, these contradict the optimality of $q$ as measured by pinball loss, which completes the proof. $\qquad\square$

### C.2   Proof of Theorem 2

*Proof.* Recall that $q$ is chosen such that:

$$q \in \arg\min_{q' \in \mathcal{H}} \mathop{\mathbb{E}}_{(x,y) \sim \mathcal{D}}[\text{PB}_{1-\alpha}(q(x), s(x,y))]$$

For point of contradiction, suppose that there is some $g \in \mathcal{G}$ and some $\alpha' \ne \alpha$ such that $\mathcal{A}_q$'s false positive rate conditional on $g(x) = 1$ is $\alpha'$. Let $\mathcal{D}_g$ be the conditional distribution on $\mathcal{D}$ conditional on $g(x) = 1$, and let $\mathcal{D}_{\bar{g}}$ be the conditional distribution on $\mathcal{D}$ conditional on $g(x) = 0$. Expanding out definitions, we have that:

$$
\begin{aligned}
\alpha' &= \mathop{\Pr}_{(x,y) \sim \mathcal{D}}[\mathcal{A}_q(x,y) = \bot | g(x) = 1] \\
&= \mathop{\Pr}_{(x,y) \sim \mathcal{D}_g}[s(x,y) \ge q(x)] \\
&= 1 - \mathop{\Pr}_{(x,y) \sim \mathcal{D}_g}[s(x,y) \le q(x)]
\end{aligned}
$$

So $\Pr_{(x,y) \sim \mathcal{D}_g}[s(x,y) \le q(x)] = 1 - \alpha'$. Let $\eta \in \mathbb{R}$ be such that $\Pr_{(x,y) \sim \mathcal{D}_g}[s(x,y) \le q(x) + \eta] = 1 - \alpha$—Note that such an $\eta$ is guaranteed to exist by continuity of the distribution on $s(x,y)$ conditional on $g(x) = 1$. Let $q'(x) = q(x) + \eta g(x)$. By Lemma 3,

$$\mathop{\mathbb{E}}_{(x,y) \sim \mathcal{D}_g}[\text{PB}_{1-\alpha}(q'(x), s(x,y))] < \mathop{\mathbb{E}}_{(x,y) \sim \mathcal{D}_g}[\text{PB}_{1-\alpha}(q(x), s(x,y))]$$

We can relate this decrease in pinball loss conditional on $g(x) = 1$ to the decrease in pinball loss on the underlying distribution $\mathcal{D}$:

$$
\begin{aligned}
&\mathop{\mathbb{E}}_{(x,y) \sim \mathcal{D}}[\text{PB}_{1-\alpha}(q'(x), s(x,y))] \\
=\ &\mathop{\Pr}_{\mathcal{D}}[g(x) = 1] \mathop{\mathbb{E}}_{(x,y) \sim \mathcal{D}_g}[\text{PB}_{1-\alpha}(q'(x), s(x,y))] + \mathop{\Pr}_{\mathcal{D}}[g(x) = 0] \mathop{\mathbb{E}}_{(x,y) \sim \mathcal{D}_{\bar{g}}}[\text{PB}_{1-\alpha}(q'(x), s(x,y))] \\
<\ &\mathop{\Pr}_{\mathcal{D}}[g(x) = 1] \mathop{\mathbb{E}}_{(x,y) \sim \mathcal{D}_g}[\text{PB}_{1-\alpha}(q'(x), s(x,y))] + \mathop{\Pr}_{\mathcal{D}}[g(x) = 0] \mathop{\mathbb{E}}_{(x,y) \sim \mathcal{D}_{\bar{g}}}[\text{PB}_{1-\alpha}(q(x), s(x,y))] \\
=\ &\mathop{\Pr}_{\mathcal{D}}[g(x) = 1] \mathop{\mathbb{E}}_{(x,y) \sim \mathcal{D}_g}[\text{PB}_{1-\alpha}(q'(x), s(x,y))] + \mathop{\Pr}_{\mathcal{D}}[g(x) = 0] \mathop{\mathbb{E}}_{(x,y) \sim \mathcal{D}_{\bar{g}}}[\text{PB}_{1-\alpha}(q'(x), s(x,y))]
\end{aligned}
$$

But because $\mathcal{H}$ is closed under additive shifts by all $g \in \mathcal{G}$, we also have that $q' \in \mathcal{H}$. Together, these contradict the optimality of $q$ as measured by pinball loss, which completes the proof. $\qquad\square$

## D   Hyperparameters

We use Ray Tune Liaw et al. [2018] for hyperparameter tuning on image datasets. All experiments use Async Hyperband Scheduler Li et al. [2020] and the Hyperopt search package Bergstra et al. [2013]. Table 4 summarizes the hyperparameters that were tuned and their configurations

Table 4: Summary of hyperparameters optimized for our quantile regressor model on all image experiments

| Hyperparameter | Configuration | Description |
|---|---|---|
| lr | loguniform($10^-6, 10^-2$) | Learning rate |
| Weight Decay | loguniform($2 * 10^-6, 5 * 10^-3$) | l2 weight regularization (excluding biases) |
| Hidden dims | choice([], [512,512]) | size and number of hidden dimensions of MLP |
| Accumulate gradient batches | choice([1, 2, 4, 8, 16, 32, 64]) | number of batches to accumulate (base batch size=32) |
| Epochs | randint(start=5, stop=50, step=5) | number of training epochs |

We ran 32 trials to find the optimal configuration per experiment. After examining the results, we found that a hidden dim of $[512, 512]$ and accumulate gradient batches of 2 (Effective batch size 64) were consistently chosen across all experiments. Similarly, epochs were mostly chosen on the $10 - 20$ range, learning rates and weight decays were consistently chosen near the middle of the value range. This indicates that hyperparameter tuning may not be especially task sensitive and can be shared across attacks.

For tabular data, we use Catboost for model training, and Optuna Akiba et al. [2019] for hyperparameter tuning. Table 5 presents the hyperparameters that were tuned and their corresponding ranges. Each model was tuned with 300 trials with 5-fold cross-validation.

Table 5: Summary of hyperparameters optimized for our quantile regressor model on tabular data

| Hyperparameter | Configuration | Description |
|---|---|---|
| depth | uniform(1,10) | Depth of a tree |
| l2_leaf_reg | loguniform(1e-2,1e+6) | Strength of L2 regularization |
| learning_rate | loguniform(1e-6,1) | Learning rate of gradient boosting |
| subsample | loguniform(1e-2,1) | Subsampling ratio at each leave node |
| iterations | loguniform(1,1000) | Number of boosting iterations |

## E   Additional Results

Here we show extended results on membership inference attacks on CIFAR-10 and CIFAR-100 for ResNet-10, -18, -34, and -50 architectures. Tables 6 and 7 show precision and pinball loss at $1\%$ and $0.1\%$ FPR for all target architectures on CIFAR-10 and CIFAR-100 respectively. We observe a strong correlation between lowest pinball loss on test samples and highest precision across the majority of experiments. Figure 5 additionally presents a visual comparison of the true positive rate and false positive rate trade-off for all tested methods on CIFAR-10 and CIFAR-100 for the ResNet-50 architecture.

Table 6: Precision and pinball loss of all membership inference attack at 1% and 0.1% false positive rates on CIFAR10 for ResNet-10, -18, -34, and -50 architectures. Pinball losses are computed on a held out test set. Lower pinball losses consistently predict better membership inference performance.

| Method | $PB_{1\%}$ | Precision @ 1% FPR | $PB_{0.1\%}$ | Precision @ 0.1% FPR |
|---|---|---|---|---|
| | | CIFAR-10 ResNet-10 | | |
| LIRA (n=2) | 0.1454 | 80.43% | 0.0194 | 89.20% |
| LIRA (n=4) | 0.1439 | 83.45% | 0.0193 | 93.15% |
| LIRA (n=6) | 0.1441 | 84.46% | 0.0193 | 94.47% |
| LIRA (n=8) | 0.1442 | 84.79% | 0.0193 | 94.67% |
| Marginal Baseline | 0.2157 | 47.84% | 0.0262 | 46.81% |
| Ours | 0.1543 | 62.69% | 0.0262 | 57.83% |
| | | CIFAR-10 ResNet-18 | | |
| LIRA (n=2) | 0.1637 | 82.17% | 0.0222 | 94.09% |
| LIRA (n=4) | 0.1607 | 84.91% | 0.0216 | 93.90% |
| LIRA (n=6) | 0.1603 | 85.33% | 0.0215 | 94.68% |
| LIRA (n=8) | 0.1603 | 85.26% | 0.0215 | 95.04% |
| Marginal Baseline | 0.1853 | 43.43% | 0.0223 | 50.00% |
| Ours | 0.1707 | 63.08% | 0.0213 | 65.65% |
| | | CIFAR-10 ResNet-34 | | |
| LIRA (n=2) | 0.1674 | 82.75% | 0.0225 | 92.45% |
| LIRA (n=4) | 0.1672 | 84.16% | 0.0223 | 94.45% |
| LIRA (n=6) | 0.1669 | 85.65% | 0.0222 | 95.40% |
| LIRA (n=8) | 0.1668 | 85.79% | 0.0222 | 95.30% |
| Marginal Baseline | 0.1893 | 51.42% | 0.0230 | 47.92% |
| Ours | 0.1743 | 73.06% | 0.0230 | 80.39% |
| | | CIFAR-10 ResNet-50 | | |
| LIRA (n=2) | 0.1913 | 78.55% | 0.0262 | 83.18% |
| LIRA (n=4) | 0.1926 | 80.52% | 0.0261 | 91.48% |
| LIRA (n=6) | 0.1930 | 83.19% | 0.0260 | 93.17% |
| LIRA (n=8) | 0.1933 | 83.00% | 0.0260 | 93.70% |
| Marginal Baseline | 0.2163 | 48.70% | 0.0287 | 60.94% |
| Ours | 0.2070 | 62.95% | 0.0277 | 59.68% |

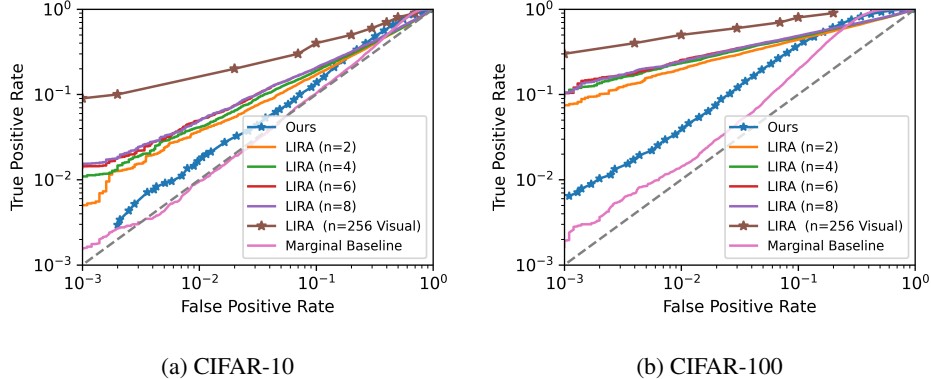

(a) CIFAR-10         (b) CIFAR-100

Figure 5: Comparing the true positive rate vs. false positive rate of our membership inference attack against the state-of-the-art shadow model approach LIRA proposed in Carlini et al. [2022] evaluated at 2, 4, 6, and 8 shadow models, and the marginal baseline proposed in Yeom et al. [2018] Our single-model quantile regression attack can reliably perform membership inference attacks on a ResNet-50 CIFAR-10/CIFAR-100 target model (91% and 68.6% test accuracies respectively) without relying on any knowledge of the target architecture. We include a visual readout of the 256-shadow model attack shown in Carlini et al. [2022] for reference. Our attack's effectiveness is dominates the marginal baseline but falls short of LIRA in this scenario. We find pinball loss to be a strong predictor of performance for membership inference attacks.

Table 7: Precision and pinball loss of all membership inference attack at 1% and 0.1% false positive rates on CIFAR100 for ResNet-10, -18, -34, and -50 architectures. Pinball losses are computed on a held out test set. Lower pinball losses consistently predict better membership inference performance.

| Method | $PB_{1\%}$ | Precision @ 1% FPR | $PB_{0.1\%}$ | Precision @ 0.1% FPR |
|---|---|---|---|---|
| | | CIFAR-100 ResNet-10 | | |
| LIRA (n=2) | 0.1486 | 95.44% | 0.0188 | 99.06% |
| LIRA (n=4) | 0.1386 | 95.95% | 0.0170 | 98.97% |
| LIRA (n=6) | 0.1385 | 96.19% | 0.0170 | 99.10% |
| LIRA (n=8) | 0.1385 | 96.14% | 0.0170 | 99.06% |
| Marginal Baseline | 0.2749 | 50.84% | 0.0390 | 47.92% |
| Ours | 0.1775 | 83.30% | 0.0390 | 77.37% |
| | | CIFAR-100 ResNet-18 | | |
| LIRA (n=2) | 0.1612 | 96.29% | 0.0193 | 99.11% |
| LIRA (n=4) | 0.1523 | 96.74% | 0.0179 | 99.46% |
| LIRA (n=6) | 0.1524 | 97.02% | 0.0181 | 99.51% |
| LIRA (n=8) | 0.1525 | 96.94% | 0.0181 | 99.48% |
| Marginal Baseline | 0.2754 | 51.71% | 0.0364 | 59.02% |
| Ours | 0.2492 | 90.37% | 0.0364 | 89.63% |
| | | CIFAR-100 ResNet-34 | | |
| LIRA (n=2) | 0.1840 | 95.73% | 0.0213 | 99.10% |
| LIRA (n=4) | 0.1732 | 96.34% | 0.0203 | 99.17% |
| LIRA (n=6) | 0.1726 | 96.45% | 0.0202 | 99.31% |
| LIRA (n=8) | 0.1727 | 96.42% | 0.0202 | 99.28% |
| Marginal Baseline | 0.2065 | 62.13% | 0.0270 | 66.67% |
| Ours | 0.1913 | 81.80% | 0.0270 | 79.64% |
| | | CIFAR-100 ResNet-50 | | |
| LIRA (n=2) | 0.1926 | 95.21% | 0.0239 | 98.65% |
| LIRA (n=4) | 0.1832 | 95.87% | 0.0227 | 98.94% |
| LIRA (n=6) | 0.1828 | 96.20% | 0.0228 | 99.02% |
| LIRA (n=8) | 0.1828 | 96.07% | 0.0228 | 98.98% |
| Marginal Baseline | 0.2188 | 58.81% | 0.0272 | 65.75% |
| Ours | 0.2006 | 79.57% | 0.0272 | 85.41% |

