# OpenReview forum: "Scalable Membership Inference Attacks via Quantile Regression"
_NeurIPS.cc/2023/Conference — NeurIPS 2023 poster_

### Official Review · Reviewer_kgzg · 2023-06-28

**Soundness:** 3 good
**Presentation:** 3 good
**Contribution:** 3 good
**Rating:** 7
**Confidence:** 4

**Summary:**

This paper introduces a new class of membership inference attacks based on performing quantile regression on the distribution of confidence scores induced by the model under attack on points that are not used in training. The approach is computationally efficient and does not require knowledge of the model's architecture, making it truly "black-box". The paper discusses the efficacy of this approach through extensive experiments on various datasets and model architectures. The experiments show that this approach is competitive with (and sometimes more effective than) much more computationally expensive shadow model approaches. Overall, the paper presents a more scalable and efficient membership inference attack.

**Strengths:**

- interesting topic
- novel method
- well-written paper


**Weaknesses:**

- more evaluation metrics needed
- compare with different attacks

**Questions:**

- I appreciate it if the authors could also compare the proposed method to the model-based attacks (https://arxiv.org/abs/1610.05820) and metric-based attacks. (https://arxiv.org/abs/2003.10595)

- Why the proposed method performs worse on C-10 and C-100 but better on IN-1k? I would suggest the authors elaborate more on it.

**Limitations:**

- Besides accessing the performance, it is also important to report the time/computational cost of different methods to further demonstrate the efficiency of the proposed method.

---

> ### Author Rebuttal · Authors · 2023-08-09
>
> Thanks for your careful reading and useful feedback --- below we address your specific questions. We're happy to further discuss any of these points if there are any remaining questions or confusions!
>
> > *I appreciate it if the authors could also compare the proposed method to the model-based attacks (https://arxiv.org/abs/1610.05820) and metric-based attacks. (https://arxiv.org/abs/2003.10595)*
>
> These are methods that pre-date LIRA, and focus on global metrics of success (rather than performance of the attacks at low FPRs). We choose LiRA as our main point of comparison as it is the state-of-the-art for membership inference when looking at performance on the presented datasets at low FPR. The original LiRA paper directly compares LiRA to the approaches you reference and shows that LiRA outperforms them, which is our basis for choosing LiRA as a state of the art method. Both methods are appropriately cited in our related work section.
>
>
>
> > *Why the proposed method performs worse on C-10 and C-100 but better on IN-1k? I would suggest the authors elaborate more on it.*
>
> We discuss this a bit in the paper, but are happy to elaborate further in the next revision. We find that uniformly across all of our experiments, the attack that obtains the lowest pinball loss has the best performance. In the smaller data regimes like C-10, we find that we can extract a quantile regression model from the LiRA shadow models that has lower pinball loss than the regression model we can obtain by directly optimizing for pinball loss (and correspondingly, out-performs). Direct optimization of pinball loss appears to work better in the large data regime, which is also where the computational benefits of our approach are most pronounced.
>
>
> > *Besides accessing the performance, it is also important to report the time/computational cost of different methods to further demonstrate the efficiency of the proposed method.*
>
> We will add final clock times to the paper in our next revision. For reference, the base model ImageNet-1k took 18 hours to train using our setup, while the quantile MIA attack took 40 hours on the same server including hyper-parameter tuning; roughly the cost of $2.2$ shadow models. On our recently added CINIC10 results, it took 85 minutes to train the base network (and each subsequent shadow model and 4 hours 40 minutes for hyper-parameter tuning; roughly the cost of $3.3$ shadow models with a performance comparable to 4 shadow models. The time cost for a single quantile regression trial was 18 minutes. On our tabular examples, we perform hyper-parameter tuning on both the base and target model and thus our approach has the same compute cost as a single shadow model.

---

> > ### Comment · Reviewer_kgzg · 2023-08-10
> >
> > The authors address my concerns, and I would be happy to raise my score.

---

> > > ### Author Response · Authors · 2023-08-10
> > >
> > > Thanks! We appreciate it.

---

### Official Review · Reviewer_648h · 2023-07-04

**Soundness:** 2 fair
**Presentation:** 3 good
**Contribution:** 2 fair
**Rating:** 5
**Confidence:** 3

**Summary:**

This work presents a novel membership inference attack that offers computational efficiency compared to state-of-the-art (SOTA) approaches. While the paper addresses a topical issue and provides an alternative attack method, there are several weaknesses that need to be addressed for a more comprehensive and convincing study, including the lack of evaluation of TPR@lowFPR, concerning empirical results, insufficient support for computational efficiency claims, and a lack of empirical evidence for the theoretical results. Addressing these weaknesses is essential to enhance the paper's credibility and provide a more comprehensive evaluation of the proposed attack method. Finally, I would be happy to increase my score if the authors could alleviate some of my concerns.

**Strengths:**

**Topical and relevant**:
The paper addresses a relevant topic in the field of membership inference attacks, which is of interest to conference readers. The proposed attack method introduces a novel approach that offers computational efficiency compared to existing SOTA methods.

**Competitive performance without training multiple shadow models**:
The authors suggest an attack method that competes with state-of-the-art approaches while being computationally more efficient. By eliminating the need to train several shadow models, their method offers a practical advantage, which is a significant contribution to the field.

**Weaknesses:**

Regarding the theoretical results:

**Lack of TPR@lowFPR evaluation**: The theoretical results presented in the paper focus on controlling the false positive rate (FPR). However, what truly matters in membership inference attacks (see [1]) is the true positive rate (TPR) at low FPR. The paper lacks analysis of TPR@lowFPR, which is crucial for assessing the attack's effectiveness. (This recent paper [2] could help the authors with this kind of analysis).

Regarding the empirical results:

**Concerning empirical results**

- The quantile attack outperforms the Likelihood Ratio Test-based Inference Attack (LIRA) in terms of TPR@lowFPR on ImageNet. This is surprising, as the Neyman-Pearson Lemma suggests that the LRT is the most powerful test. The discrepancy may be due to the limited computational resources used in training shadow models or differences in training setups when training the underlying imagenet classifier as batch size, # epochs, etc impact the succeptibility to membership inference attacks (e.g., see [1]).

- The replicated LIRA attacks with 8 and 4 shadow models outperform the read-off LIRA attack with 64 shadow models [2]. This observation questions the authors' conclusion that their attack is more favorable. It suggests that if the authors had trained 64 shadow models on ImageNet, the LIRA attack would perform better, as indicated by the evaluation on tabular datasets.

**Computational intensity claims**: The authors argue that their method is computationally more efficient by eliminating the need for training shadow models. However, there is no empirical evidence supporting this claim. It is crucial to provide comparative plots showing the tradeoff between computational cost and TPR@fixed FPR to substantiate this argument.

**Lack of empirical support for theoretical results**: The theoretical results regarding FPR control lack empirical evaluation. Conducting simulation studies or additional experiments would provide empirical evidence to support these claims.

-----
**References**

[1] Membership Inference Attacks From First Principles, https://arxiv.org/abs/2112.03570

[2] Gaussian Membership Inference Privacy, https://arxiv.org/abs/2306.07273


**Questions:**

Could the authors please elaborate on why the read-off attack results from [2] that use 64 shadow models is performing worse than the authors' implemented attacks with 4 and 8 shadow models? There must be some discrepancies in the underlying models that have been trained!?

**Limitations:**

This point should be ok.

---

> ### Author Rebuttal · Authors · 2023-08-09
>
> Thanks for your careful reading and useful feedback --- below we address your specific questions. We're happy to further discuss any of these points if there are any remaining questions or confusions!
>
>
> > *The theoretical results presented in the paper focus on controlling the false positive rate (FPR). However, what truly matters in membership inference attacks (see [1]) is the true positive rate (TPR) at low FPR. The paper lacks analysis of TPR@lowFPR, which is crucial for assessing the attack's effectiveness. (This recent paper [2] could help the authors with this kind of analysis).*
>
> Our theory concerns the false positive rate of our attacks, which is something that can be established in a model-agnostic way (i.e. without making any assumptions about the model that we are attacking). It is not possible to prove similar theorems about the attack having high true positive rates at low false positive rates, for the simple reason that not all models are vulnerable to membership inference attacks. For example, models that are trained in a differentially private way for small values of $\epsilon$ have the provable property that no membership inference attack with low false positive rate can have a high true positive rate. Thus, as with much work on hypothesis testing and membership inference, we prove one-sided theoretical guarantees, and then evaluate the tradeoff between false positive rates and true positive rates empirically. We note that our empirical analyses are indeed focused on the low false positive rate regime: e.g. the false positive rate axis in figure 1 is on a log scale, and goes as low as $10^{-4}$, and tables in our submission present precision values at low false positive rates.
>
> > *The quantile attack outperforms the Likelihood Ratio Test-based Inference Attack (LIRA) in terms of TPR@lowFPR on ImageNet....*
>
> Neyman-Pearson implies that thresholding likelihood ratios gives an optimal test _under the assumption that the likelihood ratios are known exactly_. However, in the context of membership inference attacks, these likelihood ratios are never known, as they arise from a complex empirical process. LiRA uses a simple parametric family of models to attempt to approximate the likelihoods from samples, but this approach comes with no guarantees, as in general the true likelihood ratios need not be well approximated by anything in the parametric class.
>
> Under the assumption that the likelihood ratios are monotonic in the test statistic (which is true for the parametric family of distributions used by LiRA), a likelihood ratio test also corresponds to a thresholding of the test statistic (which is what our attack does). Thus both our attack and LiRA can be viewed as consistent with the structure of an optimal hypothesis test by Neyman Pearson: we differ in what properties about the test statistic distribution we choose to estimate, and whether we perform those estimates parametrically or non parametrically. We can elaborate on this in the revision.
>
>
> > *The replicated LIRA attacks with 8 and 4 shadow models outperform the read-off LIRA attack with 64 shadow models [2]. This observation questions the authors' conclusion that their attack is more favorable. It suggests that if the authors had trained 64 shadow models on ImageNet, the LIRA attack would perform better, as indicated by the evaluation on tabular datasets.*
>
> The description in LiRA in the original paper is not completely specified, so we did our best to implement the described method; our implementation outperformed their stated results. Unfortunately, we didn't have the computational power to replicate their 64-model results (which is part of the motivation for our work, which is about reducing computational costs), but did provide their stated results for comparison. We expect that were we able to run it, our implementation of LIRA with 64 models would out-perform our implementation with 8 models, but it is not clear doing so would outperform our attack since there are diminishing marginal returns to using more models.
>
> > *Computational intensity claims: The authors argue that their method is computationally more efficient by eliminating the need for training shadow models. However, there is no empirical evidence supporting this claim. It is crucial to provide comparative plots showing the tradeoff between computational cost and TPR@fixed FPR to substantiate this argument.*
>
> We will add final clock times to the paper in our next revision. For reference, the base model ImageNet-1k took 18 hours to train using our setup, while the quantile MIA attack took 40 hours on the same server including hyper-parameter tuning; roughly the cost of $2.2$ shadow models; each individual quantile regression trial took 75 minutes. On our recently added CINIC10 results, it took 85 minutes to train the base network (and each subsequent shadow model) and 4 hours 40 minutes for hyper-parameter tuning; roughly the cost of $3.3$ shadow models with a performance comparable to 4 shadow models. The time cost for a single quantile regression trial was 18 minutes On our tabular examples, we perform hyper-parameter tuning on both the base and target model and thus our approach has the same compute cost as a single shadow model.
>
> It is worth noting that the performance of LiRA is rather sensitive to the architecture choice and the data augmentation strategies, which all are considered as hyperparameters. In the scenarios where only API access to the model is available to the attacker, then our algorithm is favorable since ours doesn't rely on any information regarding the architecture, the training process, or the augmentation strategies of the underlying model, whilst LiRA would suffer.

---

> > ### Comment · Reviewer_648h · 2023-08-15
> > **Response to Rebuttal**
> >
> > Thank you for your response and for your clarifications. I still think that the results on FPR control are not really interesting and that they do not add much insight to the paper. As the authors have partly addressed my concern, I increase my score accordingly under the assumption that the authors will include i) computational resource stats (compute + clock time as the Carlini et al attack can be parallelized and then clock time is a less reasonable measure) to all their existing results and that ii) Figure 1 from the main paper will be modified so that the readout results with $n=64$ shadow models from Carlini et al are not featured in the Figure as they seem highly misleading (authors admited that 'the description in LiRA in the original paper is not completely specified' and so comparing authors' results to those by Carlini et al in this fashion is likely to be incorrect and misleading).

---

> > > ### Author Response · Authors · 2023-08-15
> > >
> > > A sincere thanks for your response --- we really appreciate the engagement.
> > >
> > > We will update the subsequent draft with compute times, though we do highlight that we used an Async Hyper band [1] scheduler and HyperOpt Search [2] which enables parallelization of the search procedure. We will remove the 64 model readout from the figure to avoid potential confusion.
> > >
> > > [1] Li, L., Jamieson, K., Rostamizadeh, A., Gonina, E., Ben-Tzur, J., Hardt, M., ... & Talwalkar, A. (2020). A system for massively parallel hyperparameter tuning. Proceedings of Machine Learning and Systems, 2, 230-246.
> > > [2] Bergstra, J., Yamins, D., Cox, D. D. (2013) Making a Science of Model Search: Hyperparameter Optimization in Hundreds of Dimensions for Vision Architectures. TProc. of the 30th International Conference on Machine Learning (ICML 2013), June 2013, pp. I-115 to I-23.

---

### Official Review · Reviewer_5DZs · 2023-07-10

**Soundness:** 3 good
**Presentation:** 3 good
**Contribution:** 3 good
**Rating:** 7
**Confidence:** 4

**Summary:**

The paper proposes a new membership inference attack based on training an attack model with the pinball loss. This avoids the use of shadow models, a common technique for membership inference, while often outperforming attacks which do use shadow models. They evaluate their attacks on a variety of image and tabular data, and show it is competitive with or outperforms the state of the art LiRA attack on these datasets.

**Strengths:**

The proposed approach only requires the training time of a single shadow model.

The proposed attack often reaches very high precision at low false positive rates. Precision is often competitive with attacks that require many shadow models.

The evaluation considers several datasets, including both image and tabular data.

The paper is easy to read.


**Weaknesses:**

Some magic happens in the experiment section in the paragraph starting at line 290 (page 7). Here, the authors appear to introduce a different objective for their attack model, to directly learn a sample’s mean and deviation. I would like to see some ablation of the choices made in this paragraph.

The proposed attack has a higher “online” computation cost than the offline variant of LiRA. LiRA can train shadow models independently of the target model, while the quantile attack requires the target model to train its attack model.

From plots, it looks like the attack performs poorly at higher false positive rates. In settings where attack accuracy or attacker advantage is preferred to TPR and FPR, this attack might end up worse than existing attacks.

Results are only reported for a single target model, rather than averaging over multiple target models.

The attack model requires hyperparameter tuning, which can be amortized over multiple shadow models when running LiRA.

Small comment: The definition of precision should have TPR in the numerator

**Questions:**

Is the pinball loss used at all when attacking the CIFAR models? If it is, could you try to explain in more detail how it is used?

Do you have thoughts on why the attacks start to perform worse than other attacks at higher FPR?

**Limitations:**

The paper does not discuss limitations. It has minor overclaiming in some places, such as when saying the attack requires only a single model, when all results are presented after hyperparameter tuning. Another place is in claiming existing shadow model attacks require knowledge of the target model architecture, which is untrue — the LiRA paper evaluates with different architectures.

---

> ### Author Rebuttal · Authors · 2023-08-09
>
> Thanks for your careful reading and useful feedback --- below we address your specific questions. We're happy to further discuss any of these points if there are any remaining questions or confusions!
>
> > *Some magic happens in the experiment section in the paragraph starting at line 290 (page 7). Here, the authors appear to introduce a different objective for their attack model, to directly learn a sample’s mean and deviation. I would like to see some ablation of the choices made in this paragraph.*
>
> Apologies for any confusion! The general method is the same: we solve a quantile regression problem with the goal of minimizing pinball loss. We find that for smaller datasets, and simpler tasks, doing parametric quantile regression (by fitting a parametric model, rather than non-parametrically minimizing pinball loss), leads to lower pinball loss than our direct minimization used for larger datasets. This may have to do with generalization issues in these more data scarce regimes. Thanks for flagging the confusion --- we will elaborate in the next revision.
>
> > *The proposed attack has a higher “online” computation cost than the offline variant of LiRA...*
>
> It is true that to train our quantile regression model, we first need to evaluate the model we are going to attack on the points in a validation set, whereas the offline variant of LiRA does not need this evaluation step before training its shadow models. We will mention this in the next revision --- but note that this is a low-order cost compared to training LiRA's shadow models (and our own quantile regression training): Each evaluation of the model under attack requires a single forward pass. On the other hand, training a model of the same architecture requires multiple forward and backward passes per data point.
>
> > *From plots, it looks like the attack performs poorly at higher false positive rates. In settings where attack accuracy or attacker advantage is preferred to TPR and FPR, this attack might end up worse than existing attacks.*
>
> We follow the same principle in Carlini 2022 by focusing on TPR at low FPR and agree with their point that attacker advantage is a poor indicator of performance at low FPR. For reference, see the accompanying pdf with results on the CINIC10 dataset which includes attacker advantage; all tested methods perform similarly on this metric.
>
> > *Results are only reported for a single target model, rather than averaging over multiple target models.*
>
> Agreed that it would be useful to show this method works well over the randomness of SGD, if that is what you mean! We're happy to show averages/coverage intervals of this approach over that randomness in future drafts.
>
> > *Is the pinball loss used at all when attacking the CIFAR models? If it is, could you try to explain in more detail how it is used?*
>
> We found that using empirical pinball loss in hyperparameter tuning for CIFAR attacks was effective.  As mentioned above, we achieve lower test pinball loss for  small data/model regimes by using another objective function. Why this takes place is something we definitely hope to explore further.
>
> In all of our experiments, our goal is to find a quantile regression model that minimizes pinball loss. On ImageNet, we found that the most effective way to do this was to directly minimize pinball loss "non-parametrically" -- i.e. without trying to fit a parametric probability distribution to the scores. To do this, we create multiple outputs for a single quantile regression model, each with its own target quantile $\alpha$ (e.g. logarithmically spaced $\alpha$ values) and the entire network is trained to minimize pinball loss simultaneously across all outputs.
>
> For the smaller datasets like CIFAR, CINIC10, and OpenML tabular datasets, we are able to get lower test pinball loss by instead fitting a parametric Gaussian model to the score distributions;
> the model computes negative log-likelihood (NLL) of the score under the Gaussian distribution defined by the predicted mean and variance pair given a sample, and the objective function of the minimization problem is the averaged NLL over samples. After learning, based on the mean and the variance produced by the model given a sample under attack, quantiles with specific $\alpha$ values are computed from the fit Gaussian distribution.
>
> Why parametric quantile regression methods outperform direct pinball loss minimization (as measured by test pinball loss) on smaller datasets is something we hope to explore further.
>
> > *Do you have thoughts on why the attacks start to perform worse than other attacks at higher FPR?*
>
> We don't know; non-parametric quantile regression via pinball loss minimization appears to work less well (in our setting at least) for lower target quantiles. But we believe that the low false positive rate regime is the most interesting/important from the point of view of an attacker.
>
>
> > *The paper does not discuss limitations. It has minor overclaiming in some places, such as when saying the attack requires only a single model, when all results are presented after hyperparameter tuning. Another place is in claiming existing shadow model attacks require knowledge of the target model architecture, which is untrue — the LiRA paper evaluates with different architectures.*
>
> Thanks for the comments: we'll look out for instances of overclaiming and try and dial them back. We can clarify that our attack only requires a single regression model trained on a holdout $S$, though performance of the attack can be optimized using hyperparameter tuning which involves training several models and using the best one.
>
> We also highlight that difference in architectures is a sensitive parameter for shadow model attacks, as evidenced in the LiRA paper on Fig 11. Our approach doesn’t require knowledge of the architecture, since the quantile regression is essentially a different task than the original classification task.

---

> > ### Comment · Reviewer_5DZs · 2023-08-14
> > **thank you for the reply**
> >
> > I'm happy to keep my score!
> >
> > Re single target model: Generally when evaluating LiRA, it's common to train multiple models on different subsets of the training data, and verify that MI is successful for all of these. So this should capture more randomness than just SGD randomness.
> >
> > Re different ways to get low pinball loss: I think it's interesting that you may need quite different approaches to get small pinball loss on different datasets. This could be a limitation in practice, though, since this could end up with training a bunch of different models to see what works best.
> >
> > Re Fig 11 in LiRA paper: My reading of this figure is that the shadow model architecture (11a) doesn't matter that much, but I suppose that's open to interpretation.

---

> > > ### Author Response · Authors · 2023-08-15
> > >
> > > A sincere thanks for your response --- we really appreciate the engagement.
> > >
> > > We will update the draft in subsequent revisions by carrying out our attack on multiple data splits.
> > > Regarding the difficulty of directly optimizing pinball loss in small datasets. Smaller dataset sizes, task difficulty, and model expressiveness could all play an issue. So far our experiments suggests that parametric approaches are very likely to be the most successful option for this setting and we plan to investigate more on this direction.

---

### Official Review · Reviewer_NGxz · 2023-07-17

**Soundness:** 2 fair
**Presentation:** 3 good
**Contribution:** 2 fair
**Rating:** 5
**Confidence:** 3

**Summary:**

The main focus of this paper is about the membership inference attack problem, i.e., determining whether  a particular example was used in training or not. Most existing such attacks estimate the distribution of some test statistics, which are usually computationally expensive. In contrast to existing approaches, this paper proposes to perform quantile regression on the distribution of confidence scores induced by the model under attack on points that are not used in training. For implementation, the authors first collect a dataset known to be never used in training. Given a well trained model $f$, the corresponding confidence scores on collected data can be obtained. Thereafter, a quantile regression model $q$ is trained to predict the  target quantile of the data-label pair.

**Strengths:**

Reducing many (around hundred) shadow models to a single one is interesting.

Experiment studies on different image datasets, including CIFAR-10, CIFAR-100, ImageNet-1K.

**Weaknesses:**

-- Paper writing still needs some improvements. It requires efforts to follow.

-- The parameter $\alpha$ is important yet hard to adjust manually.

--  Currently, I have not observed any significant influence of the ongoing work on industry-level applications. Despite this, it is important to note that through additional analysis and exploration, there may be a possibility of uncovering substantial potential for impact in the future. As of now, the ongoing work does not appear to have made any noticeable waves in the industry. Nevertheless, by delving deeper into its intricacies and exploring its possibilities, we might unveil its capacity to bring about significant changes that could shape the industry landscape in the coming years.

-- The authors fail to provide a comprehensive explanation of the significance of membership inference attacks (MIAs). It remains unclear why it is crucial to prevent the disclosure of specific data points that were used in training a model. If we assume that we have already inferred the utilization of certain data during the training process, what kind of significant problems could arise as a result? While consulting the references cited in the submission may shed some light on these matters, the paper itself should strive to be self-contained and inclusive, providing a thorough understanding of the subject matter without relying heavily on external sources.

-- In order to facilitate better comprehension, it is necessary to reorganize the related work more effectively. By improving the organization of the related work, we can enhance its accessibility and ensure that readers can easily grasp its significance. A well-structured presentation of the related work will contribute to a clearer understanding of the subject matter.

**Questions:**

All the current MIAs, including this submission, are based on the observation that models tend to overfit their training sets. Will it still be true for current big models trained on big data? My understanding is that the assumption under MIAs is mainly caused by the generalisation ability of most models.

Will the conclusion still be true in other domains, e.g., natural language and audio?

**Limitations:**

Please refer to the weaknesses shown above.

Moreover, the contribution of this work appears to be rather limited in scope. To put it simply, the essence of this study involves training a regression model using only a subset of the available training data. It therefore makes that the significance and potential impact of this work may be somewhat restricted.

It would be better to show experiments on other image models, such as based on VIT, SwinT, BEIT, image encoder in CLIP, .etc.

---

> ### Author Rebuttal · Authors · 2023-08-09
>
> Thanks for your careful reading and useful feedback --- below we address your specific questions. We're happy to further discuss any of these points if there are any remaining questions or confusions!
>
> > *The parameter $\alpha$ is important yet hard to adjust manually.*
>
> $\alpha$ is the desired false positive rate of the attack, not a hyper-parameter that needs to be optimized. Pinball loss minimization with target quantile $1-\alpha$ explicitly produces a quantile regression model with false positive rate $\alpha$. Moreover, if we want an ensemble of attacks with multiple false positive rates, this doesn't require learning multiple quantile regression models from scratch. It can be efficiently done via multi-task learning with a shared neural network representation (and a separate ``head`` layer per target quantile).
>
> > *Currently, I have not observed any significant influence of the ongoing work on industry-level applications. Despite this, it is important to note that through additional analysis and exploration, there may be a possibility of uncovering substantial potential for impact in the future. As of now, the ongoing work does not appear to have made any noticeable waves in the industry. Nevertheless, by delving deeper into its intricacies and exploring its possibilities, we might unveil its capacity to bring about significant changes that could shape the industry landscape in the coming years.... -- The authors fail to provide a comprehensive explanation of the significance of membership inference attacks (MIAs). It remains unclear why it is crucial to prevent the disclosure of specific data points that were used in training a model. If we assume that we have already inferred the utilization of certain data during the training process, what kind of significant problems could arise as a result? While consulting the references cited in the submission may shed some light on these matters, the paper itself should strive to be self-contained and inclusive, providing a thorough understanding of the subject matter without relying heavily on external sources*
>
> Thanks for the suggestion -- we are happy to devote more space towards motivating membership inference attacks. There are several different kinds of ''privacy attacks'' on trained models, and membership inference attacks are the simplest. There are broadly two main reasons to be interested in membership inference attacks:
>
> First, membership inference attacks are building blocks that are used to launch stronger kinds of attacks, like data extraction attacks, which extract training data from the models given API access --- see e.g. Extracting Training Data from Diffusion Models, Carlini et al. 2023. Improvements in membership inference lead to improvements across the entire stack of attacks based on membership inference.
>
> Second, the guarantee of differential privacy (a strong notion of privacy that has been adopted by companies including Apple, Google, and Microsoft) is exactly that membership inference attacks can have True Positive/False positive rate curves that lie boundedly above the random guessing baseline (the diagonal on our plots). See e.g. the paper ''[Gaussian Differential Privacy](https://rss.org.uk/RSS/media/Training-and-events/Events/2020/Dong-et-al-jrssb-final.pdf)'' by Dong, Roth, and Su for an overview of the hypothesis testing view of differential privacy. So launching a successful membership inference attack falsifies a differential privacy guarantee, and is thus a form of privacy auditing that is gaining attention in industry --- see e.g. ''[Privacy Auditing with One (1) Training Run](https://arxiv.org/abs/2305.08846)'' by Steinke et al.  Improving the scalability of membership inference attacks (as our paper does) makes this form of privacy auditing more tractable.
>
> > *All the current MIAs, including this submission, are based on the observation that models tend to overfit their training sets. Will it still be true for current big models trained on big data? My understanding is that the assumption under MIAs is mainly caused by the generalisation ability of most models.*
>
> Yes: The trend we see in this work is that our attacks work better on larger training sets and larger models. This suggests that these issues may get worse, not better, in larger data and larger model regimes. Other work [2] has shown that MIAs are effective on even larger datasets and architectures than those we used here (i.e. large language models), suggesting that membership inference attacks continue to be problematic in these regimes.
>
> [2] Extracting Training Data from Large Language Models, Carlini et al. 2021
>
> > *Will the conclusion still be true in other domains, e.g., natural language and audio?*
>
> There is no reason a priori our approach would not work on these other domains, though the relevant tasks there are generally not classification. Carlini et al. 2021 and Carlini et al. 2022 both have demonstrated that it was entirely possible to attack large language models, and it is reasonable to assume that our attack would also be effective there.
>
> > *Moreover, the contribution of this work appears to be rather limited in scope. To put it simply, the essence of this study involves training a regression model using only a subset of the available training data. It therefore makes that the significance and potential impact of this work may be somewhat restricted.*
>
> We would like to clarify our regression model is trained on **validation** data, not training data, meaning this attack does not require that the attackers have access to any training data: the attack can be launched by anyone with API access to the model.

---

> > ### Comment · Reviewer_NGxz · 2023-08-18
> >
> > Thanks the author for answering my questions. Given most of them have been addressed properly, I am willing to raise my score, although I still cannot figure out significant impact of this work towards industry applications.

---

### Official Review · Reviewer_bLLW · 2023-07-19

**Soundness:** 2 fair
**Presentation:** 3 good
**Contribution:** 2 fair
**Rating:** 3
**Confidence:** 4

**Summary:**

The authors propose a novel class of membership inference attacks based on quantile regression applied to confidence score distributions. The proposed 'black-box' algorithm does not require knowledge of the model's architecture and performs competitively with state-of-the-art shadow model attacks while being computationally more efficient. The paper presents several experiments showcasing the effectiveness of the approach across various datasets.


**Strengths:**

1. The paper is well-written, with a clear and understandable motivation.

2. The technical aspects are solid and coherent.

**Weaknesses:**

1. In my opinion, the main contribution of this paper, which involves using the quantile regression model q to predict the quantiles of the score s(x,y), is not a novel idea within quantile regression. [a] has previously utilized pinball loss in conformal prediction for estimating quantiles of output. Combining quantile regression with membership inference might not present a significant technical challenge and may seem trivial.

2. The claim that the method does not require any knowledge of the model's architecture appears overstated. The method's performance heavily depends on the quantile regression model q. If q is not well-trained, the method will fail. The paper only employs the ConvNext-Tiny model in the main paper, and the sensitivity to different model architectures remains unknown.

3. I find the theoretical part confusing. In my understanding, Theorems 1 and 2 do not establish the theoretical superiority of the proposed method compared to the baseline method using constant quantiles. The paper introduces group validity, which has been extensively studied in the papers mentioned in Line 253, resulting in a lack of theoretical novelty. Additionally, even if the quantile regression model q is learned from a richer set of models, there is no concept of a 'regression model q' for the baseline method. Therefore, it is unclear how the authors concluded that the proposed method outperforms the baseline attack. It seems more intuitive to conclude that a richer model complexity for q may benefit the proposed method.

[a] Romano et al., Conformalized Quantile Regression, NeurIPS 2019

**Questions:**

1. Could you provide a sensitivity study regarding the architecture of q? Is it an important factor? Will the method perform better when q uses the same architecture as f? Will it lead to more accurate quantile estimation?

2. I mentioned my concerns about the theoretical part above.

3. The experimental results do not perform as well on small datasets. While the computation of LIRA when n=2 seems acceptable and achieves much better performance than your method, could you explain the observed gap?

**Limitations:**

I did not find the potential negative societal impact.

---

> ### Author Rebuttal · Authors · 2023-08-09
>
> Thanks for your careful reading and useful feedback — below we address your specific questions. We’re happy to further discuss any of these points if there are any remaining questions or confusions!
>
> > *In my opinion, the main contribution of this paper, which involves using the quantile regression model q to predict the quantiles of the score s(x,y), is not a novel idea within quantile regression...*
>
> Quantile regression is a classical statistical method for estimating the quantiles of a score distribution. It has been applied in many domains over the course of decades, including recently within conformal prediction, as noted by the reviewer. We do not claim to be developing novel quantile regression techniques; we use quantile regression as a tool to develop a new, scalable membership inference attack, which gives substantial improvements over the state-of-the-art on large datasets.
>
> > *The claim that the method does not require any knowledge of the model's architecture appears overstated...*
>
> We agree that this point can be elaborated on in our paper to give further substantiation. We note at the outset that the results we report use the same quantile regression architecture to attack vision models of different architectures; in contrast the results we report for shadow models use custom architectures for each model under attack. Here are the primary rationales for our claim.
> - From a theoretical perspective, because pinball loss minimization is a non-parameteric method for quantile regression, the regression function $q$ need not be from a class containing the target model (in fact, they can be completely unrelated classes). Of course the model class needs to be sufficiently expressive to obtain low loss, but this is qualitatively different than the use of shadow models, which are used to sample from the same distribution on scores as the original model did.
> - Our experiments found that large image datasets and a variety of architectures were attackable with convolutional architectures like ConvNext or ResNet; smaller, tabular datasets and models were attackable with gradient-boosting trees.
>
> As an example of architecture robustness, we were able to achieve $76.67\%$ and $ 85.46\%$ precision at $1\%$ and $0.1\%$ FPR, respectively, attacking a ResNet-50 model with a ResNet-50 quantile regression model on the CINIC10 dataset. This performance is roughly comparable to $4$ shadow models in the same conditions ($78.71\%$ and $86.99\%$ Precision respectively) with perfect knowledge of the target model's training parameters. These results are shown in more detail in the accompanying pdf.
>
> > *I find the theoretical part confusing. In my understanding, Theorems 1 and 2 do not establish the theoretical superiority of the proposed method compared to the baseline method using constant quantiles...*
>
> Although this is not how Yeom et al. describe their method, if one wants to view our attack as a generalization of Yeom et al, one can view their attack as solving a pinball loss minimization problem over the class of constant threshold models. This is because, over the class of constant functions, pinball loss is minimized at the threshold that corresponds to the target quantile of the score distribution. In this framing, our attack is a generalization in that it solves a quantile regression problem over a strictly larger model class. As we optimize over richer model classes, we get strictly stronger guarantees---the most obvious are that we find models that have lower pinball loss. As we highlight in the theory section of the paper, our guarantees are also stronger in the sense that we provide group conditional coverage--that is, the false-positive rate is no more than a target level for a large collection of subgroups. In contrast, Yeom et al. only provide a marginal guarantee that ensures a target false-positive rate over the entire distribution. Of course, the main demonstration of the superiority of our attack is given in the empirical results on large datasets.
>
> > *Could you provide a sensitivity study regarding the architecture of q?...*
>
> Our empirical results do not use the same architectures for the models we attack and our quantile regression functions. We find, empirically, that the ability of the model class class to minimize pinball loss is an excellent predictor of how well the attack will work. Just as with other learning tasks, more expressive classes will perform better if there is sufficient training data.
>
> > *The experimental results do not perform as well on small datasets...*
>
> For small datasets (and small models), you are correct that LIRA's performs quite well. As we discuss in the paper, we find that in these cases, we can extract a quantile predictor from LIRA's shadow models that has lower pinball loss than the one we train directly, and so the gap appears to be due to the difficulty of directly optimizing for pinball loss in relatively data poor settings. We note that for relatively small datasets and models, shadow model attacks like LIRA are not extremely expensive to run: the running time benefits of our approach are most apparent for learning problems corresponding to large datasets and models, which fortunately is also the setting in which our attack performs the best. This is further supported by our additional experiments on CINIC10, a generalization of the CIFAR10 dataset, where we achieve $76.67\%$ and $ 85.46\%$ precision at $1\%$ and $0.1\%$ FPR respectively against a ResNet-50 model with a ResNet-50 quantile regression model. This performance is roughly comparable to $4$ shadow models in the same conditions ($78.71\%$ and $86.99\%$ precision respectively). The computational cost of our attack, including extensive hyper-parameter tuning, is roughly $80\%$ that of the comparable shadow model attack. See the attached pdf for additional details on these results.

---

> > ### Comment · Reviewer_bLLW · 2023-08-19
> > **Thanks for the response, I am keeping my score**
> >
> > Thank you for your response. However, I'm not inclined to raise my score as my concerns remain unresolved. Although this paper doesn't center on conformal prediction, I've found that the techniques it presents align closely with the conformal prediction research line, both technically and theoretically, yielding limited novelty.
> >
> > 1. While I acknowledge that the paper doesn't introduce novel quantile regression techniques, I still perceive the amalgamation of quantile regression and the membership inference attack in this paper as straightforward. Creating a quantile regression model is fundamental in conformal prediction. Blending it with a new context doesn't inherently yield high novelty; further customization of techniques with the membership inference attack is needed.
> >
> > 2. In my view, group-conditional guarantees [1-3] have already received extensive exploration. Particularly, [1] delves into adversarial settings. The theoretical outcomes lack novelty and do not appear as solid as previous works. The theorem essentially merges the membership inference attack setting with group-conditional guarantees.
> >
> > 3. I understand that the original model and quantile regression model can belong to entirely unrelated classes. My point is that the lack of exploration into the model complexity of the quantile regression model is not justified. As you mentioned in the rebuttal, "large image datasets use ConvNext or ResNet; smaller, tabular datasets and models were attackable with gradient-boosting trees." What underlies this distinction? A sensitivity study is notably absent in both the main paper and rebuttal. The experimental results are heavily contingent on the suitable structure of the quantile regression model, demanding a strong inductive bias for its selection.
> >
> > 4. The concern about low performance on small datasets remains unresolved. LIRA's shadow models exhibit lower pinball loss than the one trained directly. What if a different quantile regression model were employed? This aligns with the issue raised in point 3.
> >
> > Given the reasons above, I'm inclined to recommend rejection for the current version of the paper.
> >
> > [1] Practical Adversarial Multivalid Conformal Prediction, NIPS 2022\
> > [2] Batch Multivalid Conformal Prediction, ICLR 2023\
> > [3] Online Multivalid Learning: Means, Moments, and Prediction Intervals, ITCS 2022

---

### Official Review · Reviewer_Fy4T · 2023-07-25

**Soundness:** 3 good
**Presentation:** 2 fair
**Contribution:** 3 good
**Rating:** 5
**Confidence:** 3

**Summary:**

This paper studies the question of membership inference attack (MIA), which can be formalized as a hypothesis testing (HT) problem.
The main contribution of this paper is introducing a new class of MIA.
The authors claim that the proposed method is competitive with SOTA MIA methods while being more computationally efficient.
The authors explain the motivation behind their method and theoretically prove the method's effectiveness (in terms of false positive rate).
This paper also conducts experiments on various datasets to evaluate their method.

**Strengths:**

1. This paper explains the relationship between MIA and HT, which is friendly to readers unfamiliar with MIA.
2. This paper compares the proposed method with those methods in Yeom et al [2018] and Carlini et al [2022]. The proposed methods seem to be a direct generalization of Yeom et al [2018].


**Weaknesses:**

My major concerns lie in the theoretical part of this paper:
1. Theorems 1 and 2 are established at the population level. The approximation error of $q$ is not discussed.
2. This paper does not include any non-asymptotic results.
2. The theoretical part of this paper only compares the proposed method with the baseline attack (Yeom et al [2018]). The difference between the proposed method and Carlini et al [2022] is not discussed.

**Questions:**

1. It seems unintuitive that the proposed method outperforms Carlini et al [2022]. Could you please explain this phenomenon?
2. Theorem 1 assumes that $\mathcal{H}$ is closed under additive shifts. Are there any examples of such hypothesis classes?

---

> ### Author Rebuttal · Authors · 2023-08-09
>
> Thanks for your careful reading and useful feedback — below we address your specific questions. We’re happy to further discuss any of these points if there are any remaining questions or confusion!
>
> > *Theorems 1 and 2 are established at the population level. The approximation error of is not discussed.*
>
> The reviewer is correct that our theorems are stated at the population level/without approximation error. That said, under the assumption that the learning of $q$ is done from some family of functions with bounded fat-shattering dimension (or other analogous quantity), standard sample complexity and uniform convergence results will imply guarantees in the finite sample setting. The reviewer is correct to note that to get worst-case theoretical bounds from finite data, one would want generalization theorems: but since the learning portion of our attack is simply solving a bounded, Lipschitz, convex optimization problem, generalization issues (and their solutions) are standard and not the focus of our work. We never rely on our theorems when evaluating our method: all results are empirical, and computed on a holdout set. The theorems are meant to give guiding intuition. We are happy to include a discussion of these points and a pointer to textbook generalization theorems for convex optimization.
>
> > *The theoretical part of this paper only compares the proposed method with the baseline attack (Yeom et al [2018]). The difference between the proposed method and Carlini et al [2022] is not discussed.*
>
> The baseline attack and our method both have guarantees that hold in the worst case, without making parametric assumptions. Carlini et al. [2022] do not have comparable theoretical results. Their approach estimates score distributions parametrically, so any comparable guarantees would require making corresponding parametric assumptions about the distribution of scores over the randomness of the dataset and model training. We can add a discussion of this point in the revision.
>
> > *It seems unintuitive that the proposed method outperforms Carlini et al [2022]. Could you please explain this phenomenon?*
>
> There are two primary reasons that our approach outperforms Carlini et al. **in some scenarios**.
>
> The first, as mentioned above, is that their method fits a simple parametric model to the scores sampled from the shadow model, and to the extent that their parametric model is ill-specified, the success of their approach will degrade; in contrast our quantile regression method (when implemented via pinball loss minimization) is non-parametric.
>
> Second, we formulate our hypothesis testing problem differently from Carlini et al., in a way that better captures the attack scenario. The hypothesis testing formulation in Carlini et al. operates on distributions of shadow models that are generated by running the training algorithm on random input datasets that purposefully include or not include the target attack example.
> Thus their hypothesis test is designed to attack a random model sampled from some specified training distribution, rather than the particular (realized) model that the membership inference attack is being launched on. In contrast, in the way we cast the hypothesis testing problem, the model under attack is fixed, and the randomness is entirely over the selection of the point under attack. Thus our hypothesis test is targeted at the specific model under attack, rather than a random model sampled from the same distribution. This better fits the actual threat model. We will add a discussion of this to the paper.
>
> > *Theorem 1 assumes that is closed under additive shifts. Are there any examples of such hypothesis classes?*
>
> Yes! Many hypothesis classes are closed under additive shifts. This includes linear and polynomial regression models, regression tree models, and any neural network architecture that has a bias/offset term. More generally, any model architecture can be made to be closed under additive shifts by adding a single additional parameter (a bias/offset term, when one does not already exist). Thus we view the additive shift assumption to be extremely mild, and easily enforceable if necessary.

---

> > ### Comment · Reviewer_Fy4T · 2023-08-16
> >
> > Thanks to the authors for their response. The rebuttal and global response have fully addressed our concerns and we have no follow-up questions. We will keep our score.

---

### Author Rebuttal · Authors · 2023-08-09

We thank the reviewers for their careful reading and useful feedback. In the attached pdf we include an additional experiment over the CINIC10 dataset to address some of the reviewers' specific concerns. We're happy to further discuss any of these points if there are any remaining questions or confusions!

---

### Decision · Program_Chairs · 2023-09-21

**Decision:**

Accept (poster)

**Comment:**

There is a consensus amongst all reviewers that this work achieves SOTA results, and has good novelties: the paper has competitive results while reducing shadow model count to a single one (reviewers NGxz, 648h, kgzg, 5DZs, bLLW).
Reviewer bLLW refers to similarity wrt. quantile regression and states that this "does not yield high novelty". Adequate references has been given by authors in the paper regarding this. Other reviewers find the paper novel enough for the MIA community.
Reviewer NGxz does not see the direct application, which is not a question particular to this work, but related to MIA in general. MIA is well established field and has applications even extending to legal domain.

Therefore the paper is recommended for acceptance.